# PFDiff: Training-Free Acceleration of Diffusion Models Combining Past and Future Scores

**Guangyi Wang**[1,2]**, Yuren Cai** [1]**, Lijiang Li**[1,2]**, Wei Peng**[3]**, Songzhi Su**[1,2]*

[1]School of Informatics, Xiamen University.
[2]Key Laboratory of Multimedia Trusted Perception and Efficient Computing,
Ministry of Education of China, Xiamen University, 361005, P.R. China.
[3]Department of Psychiatry and Behavioral Sciences, Stanford University.
{wangguangyi,caiyuren,lilijiang}@stu.xmu.edu.cn,
wepeng@stanford.edu, ssz@xmu.edu.cn

## Abstract

Diffusion Probabilistic Models (DPMs) have shown remarkable potential in image generation, but their sampling efficiency is hindered by the need for numerous denoising steps. Most existing solutions accelerate the sampling process by proposing fast ODE solvers. However, the inevitable discretization errors of the ODE solvers are significantly magnified when the number of function evaluations (NFE) is fewer. In this work, we propose *PFDiff*, a novel *training-free* and *orthogonal* timestep-skipping strategy, which enables existing fast ODE solvers to operate with fewer NFE. Specifically, PFDiff initially utilizes score replacement from past time steps to predict a "*springboard*". Subsequently, it employs this "springboard" along with foresight updates inspired by Nesterov momentum to rapidly update current intermediate states. This approach effectively reduces unnecessary NFE while correcting for discretization errors inherent in first-order ODE solvers. Experimental results demonstrate that PFDiff exhibits flexible applicability across various pre-trained DPMs, particularly excelling in conditional DPMs and surpassing previous state-of-the-art training-free methods. For instance, using DDIM as a baseline, we achieved 16.46 FID (4 NFE) compared to 138.81 FID with DDIM on ImageNet 64x64 with classifier guidance, and 13.06 FID (10 NFE) on Stable Diffusion with 7.5 guidance scale. Code is available at https://github.com/onefly123/PFDiff.

## 1 Introduction

In recent years, Diffusion Probabilistic Models (DPMs) (Sohl-Dickstein et al., 2015; Ho et al., 2020; Song et al., 2020b) have demonstrated exceptional modeling capabilities across various domains including image generation (Dhariwal & Nichol, 2021; Peebles & Xie, 2023; Karras et al., 2024), video generation (Dehghani et al., 2023), text-to-image generation (Rombach et al., 2022; Betker et al., 2023), speech synthesis (Song et al., 2022), and text-to-3D generation (Poole et al., 2022; Lin et al., 2023). They have become a key driving force advancing deep generative models. DPMs initiate with a forward process that introduces noise onto images, followed by utilizing a neural network to learn a backward process that incrementally removes noise, thereby generating images (Ho et al., 2020; Song et al., 2020b). Compared to other generative methods such as Generative Adversarial Networks (GANs) (Goodfellow et al., 2014) and Variational Autoencoders (VAEs) (Kingma & Welling, 2013), DPMs not only possess a simpler optimization target but also are capable of producing higher quality samples (Dhariwal & Nichol, 2021). However, the generation of high-quality samples via DPMs requires hundreds or thousands of denoising steps, significantly lowering their sampling efficiency and becoming a major barrier to their widespread application.

---

*Corresponding Author.

Existing techniques for rapid sampling in DPMs primarily fall into two categories. First, training-based methods (Salimans & Ho, 2022; Liu et al., 2022b; Song et al., 2023; Yin et al., 2024), which can significantly compress sampling steps, even achieving single-step sampling. However, this compression often comes with a considerable additional training cost, and these methods are challenging to apply to large pre-trained models. Second, training-free samplers (Song et al., 2020a; Lu et al., 2022a;b; Bao et al., 2022b;a; Liu et al., 2022a; Li et al., 2023; Zheng et al., 2023; Ma et al., 2024; Wimbauer et al., 2024; Zhao et al., 2023; Xue et al., 2023), which typically employ implicit or analytical solutions to Stochastic Differential Equations (SDE)/Ordinary Differential Equations (ODE) for lower-error sampling processes. For instance, Lu et al. (Lu et al., 2022a;b), by analyzing the semi-linear structure of the ODE solvers for DPMs, have sought to analytically derive optimally the solutions for DPMs' ODE solvers. These training-free sampling strategies can often be used in a plug-and-play fashion, compatible with existing pre-trained DPMs. However, when the NFE is below 10, the discretization error of these training-free methods will be significantly amplified, leading to convergence issues (Lu et al., 2022a;b), which can still be time-consuming.

To further enhance the sampling speed of DPMs, we have analyzed the potential for improvement in existing training-free accelerated methods. Initially, we observed a high similarity in the model's outputs for the existing ODE solvers when time step size $\Delta t$ is not extremely large, as illustrated in Fig. 1a. This observation led us to utilize the scores that have been computed from past time steps to approximate current scores, thereby predicting a "*springboard*". Furthermore, due to the similarities between the sampling process of DPMs and Stochastic Gradient Descent (SGD) (Robbins & Monro, 1951) as noted in Remark 1, we incorporated a *foresight* update mechanism using Nesterov momentum (Nesterov, 1983), known for accelerating SGD training. Specifically, we first predict future scores using the "springboard" to reduce errors, as shown in Fig. 1b. Then, we further replace the current scores with the future scores to facilitate a larger update step size $\Delta t$, as shown in Fig. 1c.

Motivated by these insights, we propose *PFDiff*, a timestep-skipping sampling algorithm that rapidly updates the current intermediate state combining past and future scores. Notably, PFDiff is *training-free* and *orthogonal* to existing DPMs sampling algorithms, providing a new orthogonal axis for DPMs sampling. Furthermore, we prove that PFDiff, despite utilizing fewer NFE, corrects for errors in the sampling trajectories of first-order ODE solvers, as visualized in Fig. 1c. This ensures that improving sampling speed does not compromise sampling quality; it only reduces unnecessary NFE in existing ODE solvers. To validate the orthogonality and effectiveness of PFDiff, extensive experiments were conducted on both unconditional (Ho et al., 2020; Song et al., 2020b;a) and conditional (Dhariwal & Nichol, 2021; Rombach et al., 2022) pre-trained DPMs. The results of the visualization experiment are depicted in Appendix D.9. The results indicate that PFDiff significantly enhances the sampling performance of existing ODE solvers. Particularly in conditional DPMs, PFDiff, using only DDIM as the baseline, surpasses the previous state-of-the-art training-free sampling algorithms.

## 2 BACKGROUND

### 2.1 DIFFUSION SDEs

Diffusion Probabilistic Models (DPMs) (Sohl-Dickstein et al., 2015; Ho et al., 2020; Song et al., 2020b) aim to generate $D$-dimensional random variables $x_0 \in \mathbb{R}^D$ that follow a data distribution $q(x_0)$. Taking Denoising Diffusion Probabilistic Models (DDPM) (Ho et al., 2020) as an example, these models introduce noise to the data distribution through a forward process defined over discrete time steps, gradually transforming it into a standard Gaussian distribution $x_T \sim \mathcal{N}(\mathbf{0}, \mathbf{I})$. The forward process's latent variables $\{x_t\}_{t \in [0,T]}$ are defined as follows:

$$q(x_t \mid x_0) = \mathcal{N}(x_t \mid \alpha_t x_0, \sigma_t^2 \mathbf{I}), \tag{1}$$

where $\alpha_t$ is a scalar function related to the time step $t$, with $\alpha_t^2 + \sigma_t^2 = 1$. In the model's reverse process, DDPM utilizes a neural network model $p_\theta(x_{t-1} \mid x_t)$ to approximate the transition probability $q(x_{t-1} \mid x_t, x_0)$,

$$p_\theta(x_{t-1} \mid x_t) = \mathcal{N}(x_{t-1} \mid \mu_\theta(x_t, t), \sigma_\theta^2(t)\mathbf{I}), \tag{2}$$

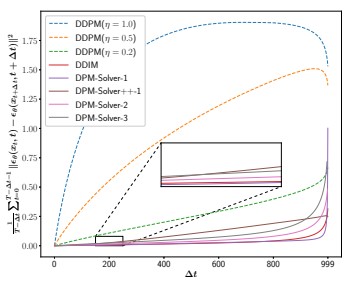
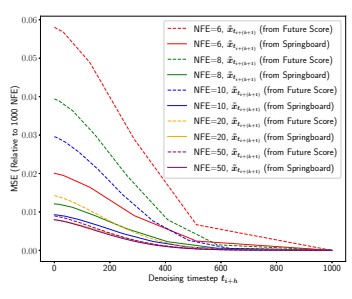
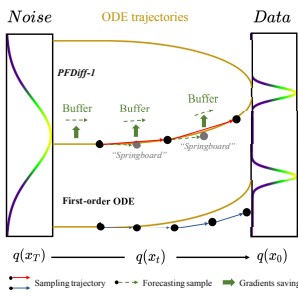

| (a) score Changes in SDE/ODE Solvers | (b) Future score More Reliable than "Springboard" | (c) Comparison of Sampling Trajectories |

Figure 1: (a) The trend of the MSE of the noise network output $\epsilon_\theta(x_t, t)$ over time step size $\Delta t$, where $\eta$ comes from $\bar{\sigma}_t$ in Eq. (6). Solid lines: ODE solvers, dashed lines: SDE solvers. (b) MSE of the status separately updated using "springboard" $\tilde{x}_{t_{i+h}}$ and future score $\epsilon_\theta(\tilde{x}_{t_{i+h}}, t_{i+h})$, relative to the sampling process with 1000 NFE, is given by: $\|\tilde{x}_{t_{i+(k+1)}} - \tilde{x}_{t_{i+(k+1)}}^{gt}\|^2$. (c) Comparison of partial sampling trajectories between PFDiff-1 and a first-order ODE solver, where the update directions are guided by the tangent direction of the sampling trajectories.

where $\sigma_\theta^2(t)$ is defined as a scalar function related to the time step $t$. By sampling from a standard Gaussian distribution and utilizing the trained neural network, samples following the data distribution $p_\theta(x_0) = \prod_{t=1}^T p_\theta(x_{t-1} \mid x_t)$ can be generated.

Furthermore, Song et al. (2020b) introduced SDE to model DPMs over continuous time steps, where the forward process is defined as:

$$\mathrm{d}x_t = f(t)x_t \mathrm{d}t + g(t)\mathrm{d}w_t, \quad x_0 \sim q(x_0), \tag{3}$$

where $w_t$ represents a standard Wiener process, and $f$ and $g$ are scalar functions of the time step $t$. It's noteworthy that the forward process in Eq. (1) is a discrete form of Eq. (3), where $f(t) = \frac{\mathrm{d}\log\alpha_t}{\mathrm{d}t}$ and $g^2(t) = \frac{\mathrm{d}\sigma_t^2}{\mathrm{d}t} - 2\frac{\mathrm{d}\log\alpha_t}{\mathrm{d}t}\sigma_t^2$. Song et al. (2020b) further demonstrated that there exists an equivalent reverse process from time step $T$ to 0 for the forward process in Eq. (3):

$$\mathrm{d}x_t = \left[f(t)x_t - g^2(t)\nabla_x \log q_t(x_t)\right]\mathrm{d}t + g(t)\mathrm{d}\bar{w}_t, \quad x_T \sim q(x_T), \tag{4}$$

where $\bar{w}$ denotes a standard Wiener process. In this reverse process, the only unknown is the *score function* $\nabla_x \log q_t(x_t)$, which can be approximated through neural networks.

## 2.2 DIFFUSION ODEs

In DPMs based on SDE, the discretization of the sampling process often requires a significant number of time steps to converge, such as the $T = 1000$ time steps used in DDPM (Ho et al., 2020). This requirement primarily stems from the randomness introduced at each time step by the SDE. To achieve a more efficient sampling process, Song et al. (2020b) utilized the Fokker-Planck equation (Øksendal & Øksendal, 2003) to derive a *probability flow ODE* related to the SDE, which possesses the same marginal distribution at any given time $t$ as the SDE. Specifically, the reverse process ODE derived from Eq. (3) can be expressed as:

$$\mathrm{d}x_t = \left[f(t)x_t - \frac{1}{2}g^2(t)\nabla_x \log q_t(x_t)\right]\mathrm{d}t, \quad x_T \sim q(x_T). \tag{5}$$

Unlike SDE, ODE avoids the introduction of randomness, thereby allowing convergence to the data distribution in fewer time steps. Song et al. (2020b) employed a high-order RK45 ODE solver (Dormand & Prince, 1980), achieving sample quality comparable to SDE at 1000 NFE with only 60 NFE. Furthermore, research such as DDIM (Song et al., 2020a) and DPM-Solver (Lu et al., 2022a) explored discrete ODE forms capable of converging in fewer NFE. For DDIM, it breaks the Markov chain constraint on the basis of DDPM, deriving a new sampling formula expressed as follows:

$$x_{t-1} = \sqrt{\alpha_{t-1}}\left(\frac{x_t - \sqrt{1-\alpha_t}\epsilon_\theta(x_t, t)}{\sqrt{\alpha_t}}\right) + \sqrt{1-\alpha_{t-1}-\bar{\sigma}_t^2}\epsilon_\theta(x_t, t) + \bar{\sigma}_t\epsilon_t, \tag{6}$$

where $\bar{\sigma}_t = \eta\sqrt{(1-\alpha_{t-1})/(1-\alpha_t)}\sqrt{1-\alpha_t/\alpha_{t-1}}$, and $\alpha_t$ corresponds to $\alpha_t^2$ in Eq. (1). When $\eta = 1$, Eq. (6) becomes a form of DDPM; when $\eta = 0$, it degenerates into an ODE, the form adopted by DDIM, which can obtain high-quality samples in fewer time steps.

**Remark 1.** *In this paper, we regard the score $\mathrm{d}\bar{x}_t$, the noise network output $\epsilon_\theta(x_t, t)$, and the score function $\nabla_x \log q_t(x_t)$ as expressing equivalent concepts. This is because Song et al. (2020b) demonstrated that $\epsilon_\theta(x_t, t) = -\sigma_t \nabla_x \log q_t(x_t)$. Moreover, we have discovered that any first-order solver of DPMs can be parameterized as $x_{t-1} = \bar{x}_t - \gamma_t \mathrm{d}\bar{x}_t + \xi\epsilon_t$. Taking DDIM (Song et al., 2020a) as an example, where $\bar{x}_t = \sqrt{\frac{\alpha_{t-1}}{\alpha_t}}x_t$, $\gamma_t = \sqrt{\frac{\alpha_{t-1}}{\alpha_t} - \alpha_{t-1}} - \sqrt{1-\alpha_{t-1}}$, $\mathrm{d}\bar{x}_t = \epsilon_\theta(x_t, t)$, and $\xi = 0$. This indicates the similarity between SGD and the sampling process of DPMs, a discovery also implicitly suggested in the research of Xue et al. (2023) and Wang et al. (2024).*

## 3 METHOD

### 3.1 SOLVING FOR REVERSE PROCESS DIFFUSION ODES

By substituting $\epsilon_\theta(x_t, t) = -\sigma_t \nabla_x \log q_t(x_t)$ (Song et al., 2020b), Eq. (5) can be rewritten as:

$$\frac{\mathrm{d}x_t}{\mathrm{d}t} = s(\epsilon_\theta(x_t, t), x_t, t) := f(t)x_t + \frac{g^2(t)}{2\sigma_t}\epsilon_\theta(x_t, t), \quad x_T \sim q(x_T). \tag{7}$$

Given an initial value $x_T$, we define the time steps $\{t_i\}_{i=0}^T$ to progressively decrease from $t_0 = T$ to $t_T = 0$. Let $\tilde{x}_{t_0} = x_T$ be the initial value. Using $T$ steps of iteration, we compute the sequence $\{\tilde{x}_{t_i}\}_{i=0}^T$ to obtain the solution of this ODE. By integrating both sides of Eq. (7), we can obtain the exact solution of this sampling ODE:

$$\tilde{x}_{t_i} = \tilde{x}_{t_{i-1}} + \int_{t_{i-1}}^{t_i} s(\epsilon_\theta(x_t, t), x_t, t)\mathrm{d}t. \tag{8}$$

For any $p$-order ODE solver, Eq. (8) can be discretely represented as:

$$\tilde{x}_{t_{i-1} \to t_i} \approx \phi(Q, \tilde{x}_{t_{i-1}}, t_{i-1}, t_i) := \tilde{x}_{t_{i-1}} + \sum_{n=0}^{p-1} h(\epsilon_\theta(\tilde{x}_{\hat{t}_n}, \hat{t}_n), \tilde{x}_{\hat{t}_n}, \hat{t}_n) \cdot \Delta\hat{t}, \quad i \in [1, \dots, T]. \tag{9}$$

Here, $Q = \left(\{\epsilon_\theta(\tilde{x}_{\hat{t}_n}, \hat{t}_n)\}_{n=0}^{p-1}, t_{i-1}, t_i\right)$ stores the set of $p$ scores computed over the intervals $t_{i-1}$ and $t_i$, where $\hat{t}_0 = t_{i-1}$, $\hat{t}_p = t_i$, and $\Delta\hat{t} = \hat{t}_{n+1} - \hat{t}_n$ denote the time step size. Particularly, when $p = 1$, $Q = \epsilon_\theta(\tilde{x}_{t_{i-1}}, t_{i-1})$. The function $\phi$ is any $p$-order ODE solver that updates the current state $\tilde{x}_{t_{i-1}}$ from time point $t_{i-1}$ to $t_i$, using the scores stored in $Q$. The function $h$ represents the way in which different $p$-order ODE solvers handle the function $s$, and its specific form depends on the solver's design. For example, in the DPM-Solver (Lu et al., 2022a), an exponential integrator is used to transform $s$ into $h$ in order to eliminate linear terms. In the case of a first-order Euler-Maruyama solver (Kloeden et al., 1992), it serves as an identity mapping of $s$.

When using the ODE solver defined in Eq. (9) for sampling, the choice of $T = 1000$ leads to significant inefficiencies in DPMs. The study on DDIM (Song et al., 2020a) first revealed that by constructing a new forward sub-state sequence of length $M + 1$ ($M \leq T$), $\{\tilde{x}_{t_i}\}_{i=0}^M$, from a subsequence of time steps $[0, \dots, T]$ and reversing this sub-state sequence, it is possible to converge to the data distribution in fewer time steps. However, as illustrated in Fig. 1a, for ODE solvers, as the time step size $\Delta t = t_i - t_{i-1}$ increases, the score direction changes slowly initially, but undergoes abrupt changes as $\Delta t \to T$. This phenomenon indicates that under minimal NFE (i.e., maximal time step size $\Delta t$) conditions, the discretization error in Eq. (9) is significantly amplified. Consequently, existing ODE solvers, when sampling under minimal NFE, must sacrifice sampling quality to gain speed, making it an extremely challenging task to reduce NFE to below 10 (Lu et al., 2022a;b). Given this, we aim to develop an efficient timestep-skipping sampling algorithm, which reduces NFE while correcting discretization errors, thereby ensuring that sampling quality is not compromised, and may even be improved.

## 3.2 SAMPLING GUIDED BY PAST SCORES

As illustrated in Fig. 1a, when the time step size $\Delta t$ (i.e., $t_i - t_{i-1}$) is not excessively large, the MSE of the noise network, defined as $\frac{1}{T-\Delta t} \sum_{t=0}^{T-\Delta t-1} \|\epsilon_\theta(x_t, t) - \epsilon_\theta(x_{t+\Delta t}, t + \Delta t)\|^2$, is remarkably similar. This phenomenon is especially pronounced in ODE-based sampling algorithms, such as DDIM (Song et al., 2020a) and DPM-Solver (Lu et al., 2022a). This observation suggests that there are many unnecessary time steps in ODE-based sampling methods during the complete sampling process (e.g., when $T = 1000$), which is one of the reasons these methods can generate samples in fewer steps. Based on this, we propose replacing the noise network of the current timestep with the output from a previous timestep to reduce unnecessary NFE without compromising the quality of the final generated samples. Specifically, for any $p$-order ODE solver $\phi$, the sampling process from $\tilde{x}_{t_{i-1}}$ to $\tilde{x}_{t_i}$ can be reformulated according to Eq. (9) as follows:

$$\tilde{x}_{t_i} \approx \phi(\{\epsilon_\theta(\tilde{x}_{\hat{t}_n}, \hat{t}_n)\}_{n=0}^{p-1}, \tilde{x}_{t_{i-1}}, t_{i-1}, t_i). \tag{10}$$

Then, we store the noise network output in a *buffer* for use in the next timestep, as follows:

$$Q \xleftarrow{\text{buffer}} \left(\{\epsilon_\theta(\tilde{x}_{\hat{t}_n}, \hat{t}_n)\}_{n=0}^{p-1}, t_{i-1}, t_i\right), \tag{11}$$

where $t_{i-1}$ and $t_i$ represent the intervals over which the set of $p$ scores are computed. For the sampling process from $\tilde{x}_{t_i}$ to $\tilde{x}_{t_{i+1}}$, we directly use the noise network output saved in the buffer from the previous timestep to replace the current timestep's noise network, thereby updating the intermediate states to the next timestep (i.e., the "springboard" $\tilde{x}_{t_{i+1}}$), as detailed below:

$$\tilde{x}_{t_{i+1}} \approx \phi(Q, \tilde{x}_{t_i}, t_i, t_{i+1}), \quad \text{where } Q = \left(\{\epsilon_\theta(\tilde{x}_{\hat{t}_n}, \hat{t}_n)\}_{n=0}^{p-1}, t_{i-1}, t_i\right). \tag{12}$$

By using this approach, we can reduce unnecessary NFE, thereby accelerating the sampling process.

**Remark 2.** *Notably, when the time step size $\Delta t$ is very large (NFE<10), the similarity between past and current scores decreases sharply, making "springboard" $\tilde{x}_{t_{i+1}}$ unreliable in Eq. (12). Therefore, in Sec. 3.3, we use $\tilde{x}_{t_{i+1}}$ solely to predict a foresight update direction (i.e., future score) to reduce errors caused by the replacement, as shown in Fig. 1b and Fig. 1c. Both past and future scores are complementary and indispensable, as demonstrated by the ablation study in Sec. 4.3.*

## 3.3 SAMPLING GUIDED BY FUTURE SCORES

As stated in Remark 1, considering the similarities between the sampling process of DPMs and SGD, and inspired by Nesterov momentum (Nesterov, 1983), we introduce a *foresight* update direction (i.e., future score) to assist the current intermediate state in achieving more efficient leapfrog updates. Notably, employing future scores is more reliable than directly using the "springboard", as discussed in Remark 2. Specifically, during the sampling process from $\tilde{x}_{t_i}$ to $\tilde{x}_{t_{i+2}}$, we consider using future scores (corresponding to time point $t_{i+1}$) to replace the current scores (corresponding to $t_i$). Continuing from Eq. (12), we estimate the future score using the "springboard" $\tilde{x}_{t_{i+1}}$ and *update* the *buffer* as follows:

$$Q \xleftarrow{\text{buffer}} \left(\{\epsilon_\theta(\tilde{x}_{\hat{t}_n}, \hat{t}_n)\}_{n=0}^{p-1}, t_{i+1}, t_{i+2}\right). \tag{13}$$

Subsequently, leveraging the concept of foresight updates, we predict a further future intermediate state $\tilde{x}_{t_{i+2}}$ using the current intermediate state $\tilde{x}_{t_i}$ along with the future score corresponding to time point $t_{i+1}$, as shown below:

$$\tilde{x}_{t_{i+2}} \approx \phi(Q, \tilde{x}_{t_i}, t_i, t_{i+2}), \quad \text{where } Q = \left(\{\epsilon_\theta(\tilde{x}_{\hat{t}_n}, \hat{t}_n)\}_{n=0}^{p-1}, t_{i+1}, t_{i+2}\right). \tag{14}$$

Furthermore, we analyze how to correct the errors of the first-order ODE solvers in the discretized Eq. (8) using future scores. Let $s_\theta(x_t, t) := s(\epsilon_\theta(x_t, t), x_t, t)$, we further analyze the term from Eq. (8) that may cause errors, $\int_{t_{i-1}}^{t_i} s_\theta(x_t, t) \mathrm{d}t$. Assuming that $s_\theta^{(n)}(x_r, r)$, $r \in [t_{i-1}, t_i]$ exists and is

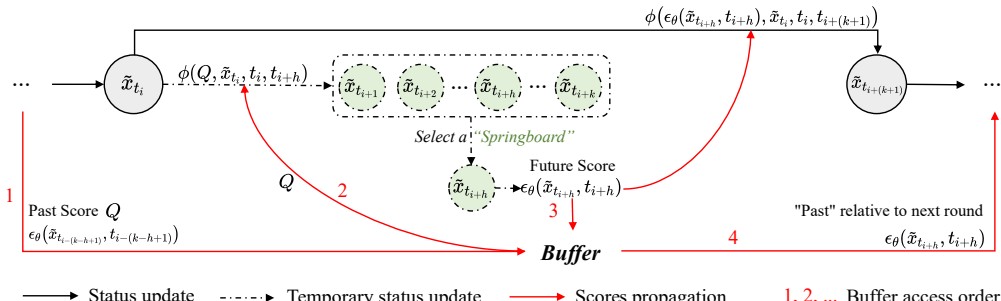

Figure 2: Illustration of a single iteration update of PFDiff-$k\_h$ combined with any first-order ODE solver $\phi$. Given specific values of $k$ and $h$ ($k \leq 3$ ($h \leq k$)), PFDiff first uses the past score $Q$ stored in the Buffer from the previous iteration to replace the current score, updating to the "springboard" $x_{t_{i+h}}$; then the future score is calculated using the "springboard"; finally, the future score is used to replace the current score, completing a full update iteration. The future score will also be passed to the next iteration as the "past" score for the next round of updates.

continuous, applying Taylor's expansion at $t = r$, we derive:

$$
\begin{aligned}
\int_{t_{i-1}}^{t_i} s_\theta\left(x_t, t\right) dt &= \int_{t_{i-1}}^{t_i} \left[\sum_{n=0}^{\infty} \frac{s_\theta^{(n)}\left(x_r, r\right)}{n!}(t - r)^n + R_n(t)\right] dt \\
&\approx \int_{t_{i-1}}^{t_i} \left[\sum_{n=0}^{\infty} \frac{s_\theta^{(n)}\left(x_r, r\right)}{n!}(t - r)^n\right] dt \\
&= \sum_{n=0}^{\infty} \frac{(t_i - r)^{n+1} - (t_{i-1} - r)^{n+1}}{(n+1)!} s_\theta^{(n)}\left(x_r, r\right) \\
&= s_\theta\left(x_r, r\right)\left(t_i - t_{i-1}\right) + \underbrace{\sum_{n=1}^{\infty} \frac{(t_i - r)^{n+1} - (t_{i-1} - r)^{n+1}}{(n+1)!} s_\theta^{(n)}\left(x_r, r\right)}_{\text{"higher-order derivative terms"}}.
\end{aligned}
\tag{15}
$$

**Proposition 3.1.** *For any given DPM first-order ODE solver, the absolute values of the coefficients for higher-order derivative terms in Eq. (15) are smaller when using the future time point $r = \varepsilon$ score compared to the current time point $r = t_{i-1}$ score, as follows (Proof in Appendix B.2):*

$$
\left|\frac{(t_i - \varepsilon)^n - (t_{i-1} - \varepsilon)^n}{n!}\right| < \left|\frac{(t_i - t_{i-1})^n}{n!}\right|, \quad \text{where } \varepsilon \in (t_{i-1}, t_i), n \geq 2.
\tag{16}
$$

Proposition 3.1 indicates that neglecting higher-order derivative terms has less impact when sampling with future scores, correcting for the discretization errors inherent in first-order ODE solvers. However, higher-order ODE solvers approximate higher-order derivative terms by estimating the noise network's output multiple times (Lu et al., 2022a;b; Zheng et al., 2023). Future scores and higher-order ODE solvers reduce the discretization errors caused by neglecting higher-order derivative terms in two parallel manners, complicating the error analysis when both methods are used simultaneously. Therefore, when using higher-order ODE solvers as a baseline, the sampling process is accelerated by only using past scores. It is only necessary to modify Eq. (14) to $\tilde{x}_{t_{i+2}} \approx \phi(Q, \tilde{x}_{t_{i+1}}, t_{i+1}, t_{i+2})$ while keeping $Q$ constant.

### 3.4 PFDIFF: SAMPLING GUIDED BY PAST AND FUTURE SCORES

Combining Sec. 3.2 and Sec. 3.3, the "springboard" $\tilde{x}_{t_{i+1}}$ obtained through Eq. (12) is used to update the buffer $Q$ in Eq. (13). In this way, we achieve our proposed efficient timestep-skipping algorithm, which we name PFDiff. Notably, during the iteration from intermediate state $\tilde{x}_{t_i}$ to $\tilde{x}_{t_{i+2}}$, we only perform a single batch computation (NFE = $p$) of the noise network in Eq. (13). Furthermore, we propose that in a single iteration process, $\tilde{x}_{t_{i+2}}$ in Eq. (14) can be modified to $\tilde{x}_{t_{i+(k+1)}}$,

achieving a $k$-step skip to sample more distant future intermediate states. Also, when $k \neq 1$, the buffer $Q$ from Eq. (13) has various computational origins. This can be accomplished by modifying "springboard" $\tilde{x}_{t_{i+1}}$ in Eq. (12) to $\tilde{x}_{t_{i+h}}$, which represents $h$ ($h \leq k$) different springboard selections. We collectively refer to this multi-step skipping and different "springboard" selection strategy as PFDiff-$k\_h$ ($h \leq k$). The algorithmic process is illustrated in Fig. 2 and Algorithm 1, with further details provided in Appendix C. Additionally, through the comparison of sampling trajectories between PFDiff-1 and a first-order ODE sampler, as shown in Fig. 1c, PFDiff-1 showcases its capability to correct the sampling trajectory of the first-order ODE sampler while reducing the NFE. Meanwhile, we observed that PFDiff completes two updates with just one score computation (1 NFE), which is equivalent to achieving an update process of a second-order ODE solver with 2 NFE. This effectiveness is derived from PFDiff's *information-efficient* update process, which utilizes both past and future scores that are complementary and indispensable. The convergence of PFDiff's sampling outcomes to the data distribution consistent with solver $\phi$ relies on the *Mean Value Theorem*, as detailed in Appendix B.3. Finally, it is important to emphasize that although PFDiff is orthogonal to an arbitrary ODE solver, PFDiff can also be viewed as an independent ODE solver, depending on the perspective.

## 3.5 ANALYSIS OF EFFECTIVENESS BASED ON THE SHAPE OF THE TRAJECTORY

In Proposition 3.1, we theoretically analyze how PFDiff corrects the error of the first-order ODE solver to achieve efficient sampling. In this section, we explain the effectiveness of PFDiff from the perspective of the trajectory's geometric shape. Previous studies have explored the sampling trajectories of diffusion models (Sabour et al., 2024; Zhou et al., 2024; Chen et al., 2024). Zhou et al. (2024) pointed out that the sampling trajectories of DPMs lie in a low-dimensional subspace embedded in a high-dimensional space, and the trajectory shapes closely resemble a straight line. This finding supports the strategy of using past scores to replace the current score in PFDiff as reliable. Moreover, Chen et al. (2024) further noted that the sampling trajectories exhibit a "boomerang" shape, meaning the curvature of the sampling trajectory starts small, then increases, and finally decreases. Based on this observation, we can analyze that the first-order ODE solver, which samples along the tangent direction, leads to larger discretization errors in regions of the trajectory with large curvature. On the other hand, PFDiff uses future scores to predict the future update direction, thereby correcting the discretization errors introduced by sampling along the tangent direction. In Fig. 1c, we vividly demonstrate the sampling correction process of PFDiff for first-order ODE solvers, thereby validating the effectiveness of PFDiff.

---

**Algorithm 1** PFDiff-1

---

**Require:** initial value $x_T$, NFE $N$, model $\epsilon_\theta$, any $p$-order solver $\phi$
1: Define time steps $\{t_i\}_{i=0}^M$ with $M = 2N - 1p$
2: $\tilde{x}_{t_0} \leftarrow x_T$
3: $Q \xleftarrow{\text{buffer}} \left( \{\epsilon_\theta(\tilde{x}_{\hat{t}_n}, \hat{t}_n)\}_{n=0}^{p-1}, t_0, t_1 \right)$          ▷ Initialize buffer
4: $\tilde{x}_{t_1} = \phi(Q, \tilde{x}_{t_0}, t_0, t_1)$
5: **for** $i \leftarrow 1$ to $\frac{M}{p} - 2$ **do**
6:   **if** $(i - 1) \mod 2 = 0$ **then**
7:    $\tilde{x}_{t_{i+1}} = \phi(Q, \tilde{x}_{t_i}, t_i, t_{i+1})$       ▷ Updating guided by past scores
8:    $Q \xleftarrow{\text{buffer}} \left( \{\epsilon_\theta(\tilde{x}_{\hat{t}_n}, \hat{t}_n)\}_{n=0}^{p-1}, t_{i+1}, t_{i+2} \right)$    ▷ Update buffer (overwrite)
9:    **if** $p = 1$ **then**
10:     $\tilde{x}_{t_{i+2}} = \phi(Q, \tilde{x}_{t_i}, t_i, t_{i+2})$    ▷ Anticipatory updating guided by future scores
11:    **else if** $p > 1$ **then**
12:     $\tilde{x}_{t_{i+2}} = \phi(Q, \tilde{x}_{t_{i+1}}, t_{i+1}, t_{i+2})$    ▷ The higher-order solver uses only past scores
13:    **end if**
14:   **end if**
15: **end for**
16: **return** $\tilde{x}_{t_M}$

---

# 4 EXPERIMENTS

In this section, we validate the effectiveness of PFDiff as an *orthogonal* and *training-free* sampler through a series of extensive experiments (**the results of the visualization experiment are shown in Figs. 7-13 of Appendix D.9**). This sampler can be integrated with any order of ODE solvers, thereby significantly enhancing the sampling efficiency of various types of pre-trained DPMs. To systematically showcase the performance of PFDiff, we categorize the pre-trained DPMs into two main types: conditional and unconditional. Unconditional DPMs are further subdivided into discrete and continuous, while conditional DPMs are subdivided into classifier guidance and classifier-free guidance. In choosing ODE solvers, we utilized the widely recognized first-order DDIM (Song et al., 2020a), Analytic-DDIM (Bao et al., 2022b), and the higher-order DPM-Solver (Lu et al., 2022a) as baselines. For each experiment, we use the Fréchet Inception Distance (FID↓) (Heusel et al., 2017) as the primary evaluation metric, and provide the experimental results of the Inception Score (IS↑) (Salimans et al., 2016) in the Appendix D.7 for reference. Lastly, apart from the ablation studies on parameters $k$ and $h$ discussed in Sec. 4.3, we showcase the optimal results of PFDiff-$k\_h$ (where $k = 1, 2, 3$ and $h \leq k$) across six configurations as a performance demonstration of PFDiff. As described in Appendix C, this does not increase the computational burden in practical applications. All experiments were conducted on an NVIDIA RTX 3090 GPU.

## 4.1 UNCONDITIONAL SAMPLING

For unconditional DPMs, we selected discrete DDPM (Ho et al., 2020) and DDIM (Song et al., 2020a), as well as pre-trained models from continuous ScoreSDE (Song et al., 2020b), to assess the effectiveness of PFDiff. For these pre-trained models, all experiments sampled 50k samples to compute evaluation metrics.

For unconditional discrete DPMs, we first select first-order ODE solvers DDIM (Song et al., 2020a) and Analytic-DDIM (Bao et al., 2022b) as baselines, while implementing SDE-based DDPM (Ho et al., 2020) and Analytic-DDPM (Bao et al., 2022b) methods for comparison, where $\eta = 1.0$ is from $\bar{\sigma}_t$ in Eq. (6). We conduct experiments on the CIFAR10 (Krizhevsky et al., 2009) and CelebA 64x64 (Liu et al., 2015) datasets using the quadratic time steps employed by DDIM. By varying the NFE from 6 to 20, the evaluation metric FID↓ is shown in Figs. 3a and 3b. Additionally, experiments with uniform time steps are conducted on the CelebA 64x64, LSUN-bedroom 256x256 (Yu et al., 2015), and LSUN-church 256x256 (Yu et al., 2015) datasets, with more results available in Appendix D.2. Our experimental results demonstrate that PFDiff, based on pre-trained models of discrete unconditional DPMs, significantly improves the sampling efficiency of DDIM and Analytic-DDIM samplers across multiple datasets. For instance, on the CIFAR10 dataset, PFDiff combined with DDIM achieves a FID of 4.10 with only 15 NFE, comparable to DDIM's performance of 4.04 FID with 1000 NFE. This is something other time-step skipping algorithms (Bao et al., 2022b; Ma et al., 2024) that sacrifice sampling quality for speed cannot achieve. Furthermore, in Appendix D.2, by varying $\eta$ from 1.0 to 0.0 in Eq. (6) to control the scale of noise introduced by SDE, we observe that as $\eta$ decreases (reducing noise introduction), the performance of PFDiff gradually improves. This is consistent with the trend shown in Fig. 1a, where reducing noise introduction leads to an improvement in the similarity of the model's outputs.

For unconditional continuous DPMs, we choose the DPM-Solver-1, -2 and -3 (Lu et al., 2022a) as the baseline to verify the effectiveness of PFDiff as an orthogonal timestep-skipping algorithm on the first and higher-order ODE solvers. We conducted experiments on the CIFAR10 (Krizhevsky et al., 2009) using quadratic time steps, varying the NFE. The experimental results using FID↓ as the evaluation metric are shown in Fig. 3c. More experimental details can be found in Appendix D.3. We observe that PFDiff consistently improves the sampling performance over the baseline with fewer NFE settings, particularly in cases where higher-order ODE solvers fail to converge with a small NFE (below 10) (Lu et al., 2022a).

## 4.2 CONDITIONAL SAMPLING

For conditional DPMs, we selected the pre-trained models of the widely recognized classifier guidance paradigm, ADM-G (Dhariwal & Nichol, 2021), and the classifier-free guidance paradigm, Stable-Diffusion (Rombach et al., 2022), to validate the effectiveness of PFDiff. We employed

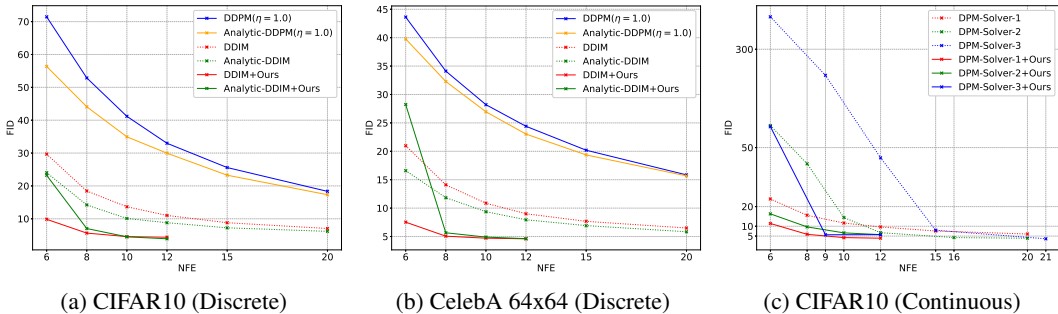

Figure 3: Unconditional sampling results. We report the FID↓ for different methods by varying the number of function evaluations (NFE), evaluated on 50k samples.

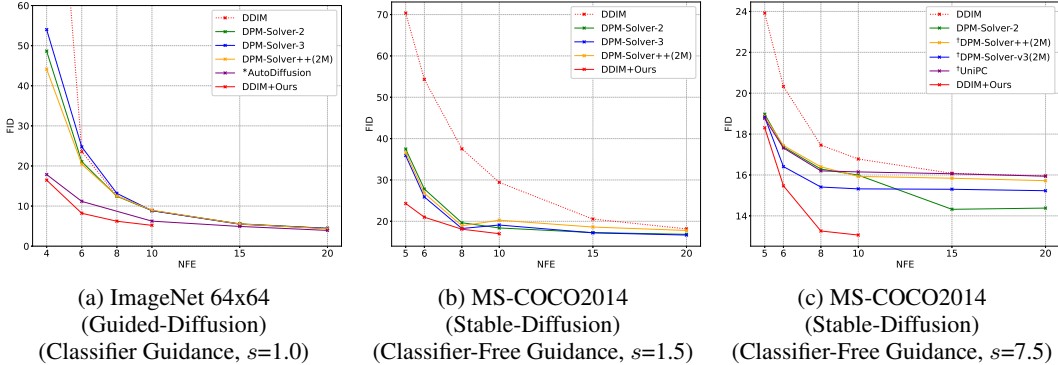

Figure 4: Conditional sampling results. We report the FID↓ for different methods by varying the NFE. Evaluated: ImageNet 64x64 with 50k, others with 10k samples. *AutoDiffusion (Li et al., 2023) requires additional search costs. †We borrow the results reported in DPM-Solver-v3 (Zheng et al., 2023) directly.

uniform time steps setting and the DDIM (Song et al., 2020a) ODE solver as a baseline across all datasets. Evaluation metrics were computed by sampling 50k samples on the ImageNet 64x64 (Deng et al., 2009) dataset for ADM-G and 10k samples on other datasets, including ImageNet 256x256 in ADM-G and MS-COCO2014 (Lin et al., 2014) in Stable-Diffusion.

For conditional DPMs employing the classifier guidance paradigm, we conducted experiments on the ImageNet 64x64 dataset with a guidance scale ($s$) set to 1.0. For comparison, we implemented DPM-Solver-2 and -3 (Lu et al., 2022a), and DPM-Solver++(2M) (Lu et al., 2022b), which exhibit the best performance on conditional DPMs. Additionally, we introduced the AutoDiffusion method (Li et al., 2023) using DDIM as a baseline for comparison, noting that this method incurs additional search costs. We compared FID↓ scores by varying the NFE as depicted in Fig. 4a, with corresponding visual comparisons shown in Fig. 7b. We observed that PFDiff reduced the FID from 138.81 with 4 NFE in DDIM to 16.46, achieving an 88.14% improvement in quality. The visual results in Fig. 7b further demonstrate that, at the same NFE setting, PFDiff achieves higher-quality sampling. Furthermore, we evaluated PFDiff's sampling performance based on DDIM on the large-scale ImageNet 256x256 dataset. Detailed results are provided in Appendix D.4.

For conditional, classifier-free guidance paradigms of DPMs, we employed the `sd-v1-4` checkpoint and computed the FID↓ scores on the validation set of MS-COCO2014 (Lin et al., 2014). We conducted experiments with a guidance scale ($s$) set to 7.5 and 1.5. For comparison, we implemented DPM-Solver-2 and -3 (Lu et al., 2022a), and DPM-Solver++(2M) (Lu et al., 2022b) methods. At $s = 7.5$, we introduced the state-of-the-art method reported in DPM-Solver-v3 (Zheng et al., 2023) for comparison, along with DPM-Solver++(2M) (Lu et al., 2022b), UniPC (Zhao et al., 2023), and DPM-Solver-v3(2M). The FID↓ metrics by varying the NFE are presented in Figs. 4b and 4c, with additional visual results illustrated in Fig. 7a. We observed that PFDiff, solely based on DDIM, achieved state-of-the-art results during the sampling process of Stable-Diffusion, thus demonstrat-

Table 1: Sample quality measured by FID↓ on the MS-COCO2014 dataset (Lin et al., 2014), using Stable-Diffusion (Rombach et al., 2022) pre-trained model with a guidance scale of 7.5, varying the number of function evaluations (NFE). Evaluated with 10k samples.

| Method | NFE | | | | | |
|---|---|---|---|---|---|---|
| | 4 | 6 | 8 | 10 | 15 | 20 |
| DPM-Solver-1 (Lu et al., 2022a) | 35.48 | 20.33 | 17.46 | 16.78 | 16.08 | 15.95 |
| + PFDiff | **29.02** | **15.47** | **13.26** | **13.06** | **13.57** | **13.97** |
| DPM-Solver-2 (Lu et al., 2022a) | 184.21 | 157.95 | 148.67 | 135.81 | 92.62 | 40.47 |
| + PFDiff | **147.20** | **106.24** | **57.07** | **31.66** | **17.87** | **14.13** |

ing the efficacy of PFDiff. Further experimental details can be found in Appendix D.5. Additionally, to further validate the orthogonality of PFDiff, we conducted experiments on the original (single-step) DPM-Solver-1 and -2, comparing the performance with and without the PFDiff, using the Stable-Diffusion pre-trained model, as shown in Tab. 1. The experimental results demonstrate that PFDiff effectively enhances the performance of DPM-Solver across different orders.

### 4.3 ABLATION STUDY

We conducted ablation experiments on the six different algorithm configurations of PFDiff mentioned in Appendix C, with $k = 1, 2, 3$ ($h \leq k$). Specifically, we evaluated the FID↓ scores on the unconditional and conditional pre-trained DPMs (Ho et al., 2020; Dhariwal & Nichol, 2021). Detailed experimental setups and results can be found in Appendix D.6.1. The experimental results indicate that for various pre-trained DPMs, the choice of parameters $k$ and $h$ is not critical, as most combinations of $k$ and $h$ within PFDiff can enhance the sampling efficiency over the baseline. Moreover, with $k = 2$ and $h = 1$ fixed, PFDiff-2_1 can always improve the baseline's sampling quality within the range of 4∼20 NFE. For even better sampling quality, one can sample a small subset of examples (e.g., 5k) to compute evaluation metrics or directly conduct visual analysis, easily identifying the most effective $k$ and $h$ combinations. Furthermore, in Appendix D.6.1, we propose an automatic search strategy with almost no additional cost, which can more rapidly obtain more competitive combinations of $k$ and $h$ based on truncation error.

To validate the effectiveness of PFDiff, a key factor is its information-efficient update process, which uses both past and future scores that are complementary and indispensable in jointly guiding first-order ODE solvers. We employ DDIM (Song et al., 2020a) as the baseline, removing past and future scores separately. Moreover, we introduce methods (Ma et al., 2024) that cache part of past scores for comparison. As shown in Appendix D.6.2, experimental results indicate that only past (including cache) or only future scores can slightly improve sampling performance, but their combination (i.e., the complete PFDiff) significantly enhances the performance of first-order ODE solvers, especially with very few NFE (<10). Additionally, we provide experimental results on inference time in Appendix D.6.2, revealing that methods (Ma et al., 2024) that cache part of past scores not only incur additional inference costs but also exhibit relatively weak acceleration effects with few NFE (<10). However, PFDiff and the used baseline have consistent inference times and exhibit significantly accelerated effects, further validating its effectiveness.

## 5 CONCLUSION

In this paper, based on the recognition that the ODE solvers of DPMs exhibit significant similarity in model outputs when the time step size is not excessively large, and with the aid of a foresight update mechanism, we propose PFDiff, a novel method that leverages past and future scores to rapidly update the current intermediate state. This approach effectively reduces the unnecessary number of function evaluations (NFE) in the ODE solvers and significantly corrects the errors of first-order ODE solvers during the sampling process. Extensive experiments demonstrate the orthogonality and effectiveness of PFDiff on both unconditional and conditional pre-trained DPMs, especially in conditional pre-trained DPMs where PFDiff outperforms previous state-of-the-art training-free sampling methods.

ETHICS STATEMENT

DPMs, like GANs and VAEs, may be utilized as deep generative models for generating fake and malicious content. The proposed PFDiff can accelerate the generation of DPMs, which may facilitate the rapid creation of such content, thereby posing a potential negative impact on society.

REPRODUCIBILITY STATEMENT

Our code is based on the official implementations of DDIM (Song et al., 2020a), DPM-Solver (Lu et al., 2022a), and Analytic-DPM (Bao et al., 2022b). We utilized unconditional checkpoints from DDPM (Ho et al., 2020), DDIM (Song et al., 2020a), and ScoreSDE (Song et al., 2020b), as well as conditional checkpoints from AMD-G (Dhariwal & Nichol, 2021) and Stable-Diffusion (Rombach et al., 2022). Detailed experimental settings and algorithm implementations are described in Appendices C and D.

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

## A    RELATED WORK

While the solvers for Diffusion Probabilistic Models (DPMs) are categorized into two types, SDE and ODE, most current accelerated sampling techniques are based on ODE solvers due to the observation that the stochastic noise introduced by SDE solvers hampers rapid convergence. ODE-based solvers are further divided into training-based methods (Salimans & Ho, 2022; Liu et al., 2022b; Song et al., 2023; Yin et al., 2024) and training-free samplers (Song et al., 2020a; Lu et al., 2022a;b; Bao et al., 2022b;a; Liu et al., 2022a; Li et al., 2023; Zheng et al., 2023; Ma et al., 2024; Wimbauer et al., 2024; Zhao et al., 2023; Xue et al., 2023). Training-based methods can notably reduce the number of sampling steps required for DPMs. An example of such a method is the knowledge distillation algorithm proposed by Song et al. (2023), which achieves one-step sampling for DPMs. This sampling speed is comparable to that of GANs (Goodfellow et al., 2014) and VAEs (Kingma & Welling, 2013). However, these methods often entail significant additional costs for distillation training. This requirement poses a challenge when applying them to large pre-trained DPM models. Therefore, our work primarily focuses on training-free, ODE-based accelerated sampling strategies.

Training-free accelerated sampling techniques based on ODE can generally be applied in a plug-and-play manner, adapting to existing pre-trained DPMs. These methods can be categorized based on the order of the ODE solver—that is, the NFE required per sampling iteration—into first-order (Song et al., 2020a; Bao et al., 2022b;a; Liu et al., 2022a) and higher-order (Lu et al., 2022a;b; Zheng et al., 2023; Zhao et al., 2023; Dormand & Prince, 1980). Typically, higher-order ODE solvers tend to sample at a faster rate but may fail to converge when the NFE is low (below 10), sometimes performing even worse than first-order ODE solvers. In addition, there are orthogonal techniques for accelerated sampling. For instance, Li et al. (2023) build upon existing ODE solvers and use search algorithms to find optimal sampling sub-sequences and model structures to further speed up the sampling process; Ma et al. (2024) and Wimbauer et al. (2024) observe that the low-level features of noise networks at adjacent time steps exhibit similarities, and they use caching techniques to substitute some of the network's low-level features, thereby further reducing the number of required time steps.

The algorithm we propose belongs to the class of training-free and orthogonal accelerated sampling techniques, capable of further accelerating the sampling process on the basis of existing first-order and higher-order ODE solvers. Compared to the aforementioned orthogonal sampling techniques, even though the skipping strategy proposed by Ma et al. (2024) and Wimbauer et al. (2024) effectively accelerates the sampling process, it may do so at the cost of reduced sampling quality, making it challenging to reduce the NFE below 50. Although Li et al. (2023) can identify more optimal subsampling sequences and model structures, this implies higher search costs. In contrast, our proposed orthogonal sampling algorithm is more efficient in skipping time steps. First, our skipping strategy does not require extensive search costs. Second, we can correct the sampling trajectory of first-order ODE solvers while reducing the number of sampling steps required by existing ODE solvers, achieving more efficient accelerated sampling.

## B    PROOF OF CONVERGENCE AND ERROR CORRECTION FOR PFDIFF

In this section, we first prove that neglecting higher-order derivative terms has a smaller impact on the first-ODE solvers when using future scores (i.e., Proposition 3.1). Subsequently, we prove the convergence of PFDiff based on the *Mean Value Theorem* for Integrals.

### B.1    ASSUMPTIONS

For PFDiff-$k\_h$ we make the following assumptions:

**Assumption B.1.** *The higher-order derivatives $s_\theta^{(n)}(x_r, r)$ (as a function of $r$), as defined in Eq. (15), where $r \in [t_{i-1}, t_i]$ and $n \geq 1$, exist and are continuous.*

**Assumption B.2.** *When the time step size $\Delta t = t_i - t_{i-(k-h+1)}$ is not excessively large, the output estimates of the noise network based on the $p$-order ODE solver at different time steps are approximately the same, that is, $\left( \{\epsilon_\theta(\tilde{x}_{\hat{t}_n}, \hat{t}_n)\}_{n=0}^{p-1}, t_i, t_{i+h} \right) \approx \left( \{\epsilon_\theta(\tilde{x}_{\hat{t}_n}, \hat{t}_n)\}_{n=0}^{p-1}, t_{i-(k-h+1)}, t_i \right).$*

**Assumption B.3.** *For the integral from time step $t_i$ to $t_{i+(k+1)}$, $\int_{t_i}^{t_{i+(k+1)}} s(\epsilon_\theta(x_t, t), x_t, t)\mathrm{d}t$, there exist intermediate time steps $t_{\tilde{s}}, t_s \in [t_i, t_{i+(k+1)}]$ such that $\int_{t_i}^{t_{i+(k+1)}} s(\epsilon_\theta(x_t, t), x_t, t)\mathrm{d}t = s(\epsilon_\theta(x_{t_{\tilde{s}}}, t_{\tilde{s}}), x_{t_{\tilde{s}}}, t_{\tilde{s}}) \cdot (t_{i+(k+1)} - t_i) = h(\epsilon_\theta(x_{t_s}, t_s), x_{t_s}, t_s) \cdot (t_{i+(k+1)} - t_i)$ holds, where the definition of the function $h$ remains consistent with Sec. 3.1.*

The first assumption ensures the application of Taylor expansion in Eq. (15). The second assumption is based on the observation in Fig. 1a that when $\Delta t$ is not excessively large, the MSE of the noise network remains almost unchanged across different time steps. The last one is based on the *Mean Value Theorem* for Integrals, which states that if $f(x)$ is a continuous real-valued function on a closed interval $[a, b]$, then there exists at least one point $c \in [a, b]$ such that $\int_a^b f(x)\mathrm{d}x = f(c)(b-a)$ holds.

**Remark 3.** *It is important to note that the Mean Value Theorem for Integrals originally applies to real-valued functions and does not directly apply to vector-valued functions (Cheney et al., 2001). However, the study by Zhou et al. (2024), which uses Principal Component Analysis (PCA) on the trajectories of the ODE solvers for DPMs, demonstrates that these trajectories almost lie in a two-dimensional plane. The finding ensures the applicability of the Mean Value Theorem for Integrals in Assumption B.3.*

### B.2 PROOF OF PROPOSITION 3.1

In this section, we prove that Eq. (16) holds, where $\varepsilon \in (t_{i-1}, t_i)$ and $n \geq 2$. First, given $\varepsilon \in (t_{i-1}, t_i)$, we have:

$$|t_i - \varepsilon| + |t_{i-1} - \varepsilon| = |t_i - t_{i-1}|. \tag{B.1}$$

Next, we analyze the Eq. (16) based on the parity of $n$.

**When $n$ is even ($n \geq 2$):** We derive:

$$\left| \frac{(t_i - \varepsilon)^n - (t_{i-1} - \varepsilon)^n}{n!} \right| = \left| \frac{|t_i - \varepsilon|^n - |t_{i-1} - \varepsilon|^n}{n!} \right|$$
$$< \max \left( \frac{|t_i - \varepsilon|^n}{n!}, \frac{|t_{i-1} - \varepsilon|^n}{n!} \right) \tag{B.2}$$
$$< \frac{|t_i - t_{i-1}|^n}{n!} = \left| \frac{(t_i - t_{i-1})^n}{n!} \right|.$$

Here, the second "$<$" holds because: due to $\varepsilon \in (t_{i-1}, t_i)$ and Eq. (B.1), we have $|t_i - \varepsilon| < |t_i - t_{i-1}|$ and $|t_{i-1} - \varepsilon| < |t_i - t_{i-1}|$, thus validating the second "$<$".

**When $n$ is odd ($n \geq 3$):** Since $\varepsilon \in (t_{i-1}, t_i)$, if $t_i - \varepsilon > 0$, then $t_{i-1} - \varepsilon < 0$; if $t_i - \varepsilon < 0$, then $t_{i-1} - \varepsilon > 0$. Therefore, we obtain:

$$\left| \frac{(t_i - \varepsilon)^n - (t_{i-1} - \varepsilon)^n}{n!} \right| = \frac{|t_i - \varepsilon|^n + |t_{i-1} - \varepsilon|^n}{n!}. \tag{B.3}$$

Let $a = |t_i - \varepsilon|$, $b = |t_{i-1} - \varepsilon|$, and $c = |t_i - t_{i-1}|$; we have $a, b, c > 0$ and $c > a, b$. Next, using mathematical induction, we prove $a^n + b^n < c^n$, where $n \geq 3$ and $a + b = c$ (Eq. (B.1)).

- When $n = 3$, we have:
$$c^3 = (a+b)^3 = a^3 + 3a^2b + 3ab^2 + b^3 > a^3 + b^3, \tag{B.4}$$
  which holds.
- When $n = k$ ($k \geq 3$, $k \in \mathbb{N}$), suppose $a \leq b$, then $a^k + b^k < c^k$ holds.
- When $n = k + 1$, we have:
$$a^{k+1} + b^{k+1} = a \cdot a^k + b \cdot b^k \leq b \cdot a^k + b \cdot b^k = b \cdot (a^k + b^k) < b \cdot c^k < c^{k+1}, \tag{B.5}$$
  which holds.

Thus, $a^n + b^n < c^n$ holds, where $n \geq 3$ and $a + b = c$. Furthermore, we obtain:

$$\frac{|t_i - \varepsilon|^n + |t_{i-1} - \varepsilon|^n}{n!} < \frac{|t_i - t_{i-1}|^n}{n!} = \left| \frac{(t_i - t_{i-1})^n}{n!} \right|. \tag{B.6}$$

In conclusion, by combining Eq. (B.2), Eq. (B.3), and Eq. (B.6), we have proven Eq. (16).

### B.3 PROOF OF CONVERGENCE FOR PFDIFF

**Assumption B.2 ensures the convergence of PFDiff-$k\_h$ using past scores.** Starting from Eq. (8), we consider an iteration process of a $p$-order ODE solver from $\tilde{x}_{t_i}$ to $\tilde{x}_{t_{i+h}}$, where $h$ is the "*springboard*" choice determined by PFDiff-$k\_h$. This iterative process can be expressed as:

$$\tilde{x}_{t_{i+h}} = \tilde{x}_{t_i} + \int_{t_i}^{t_{i+h}} s(\epsilon_\theta(x_t,t), x_t, t)\mathrm{d}t. \tag{B.7}$$

Discretizing Eq. (B.7) yields:

$$\begin{aligned}
\tilde{x}_{t_i \to t_{i+h}} &\approx \tilde{x}_{t_i} + \sum_{n=0}^{p-1} h(\epsilon_\theta(\tilde{x}_{\hat{t}_n}, \hat{t}_n), \tilde{x}_{\hat{t}_n}, \hat{t}_n) \cdot (\hat{t}_{n+1} - \hat{t}_n) \\
&= \tilde{x}_{t_i} + \sum_{n=i}^{i+h-1} h(\epsilon_\theta(\tilde{x}_{t_n}, t_n), \tilde{x}_{t_n}, t_n) \cdot (t_{n+1} - t_n),
\end{aligned} \tag{B.8}$$

where the function $h$ represents the different solution methodologies applied by various $p$-order ODE solvers to the function $s$, consistent with Sec. 3.1. To accelerate sampler convergence and reduce unnecessary NFE, we adopt Assumption B.2, namely guiding the sampling of the current intermediate state by utilizing past score information. Specifically, we approximate that $\left(\{\epsilon_\theta(\tilde{x}_{\hat{t}_n}, \hat{t}_n)\}_{n=0}^{p-1}, t_i, t_{i+h}\right) \approx \left(\{\epsilon_\theta(\tilde{x}_{\hat{t}_n}, \hat{t}_n)\}_{n=0}^{p-1}, t_{i-(k-h+1)}, t_i\right)$, where $k$ represents the number of steps skipped in one iteration by PFDiff-$k\_h$. Eq. (B.8) can be further rewritten as:

$$\begin{aligned}
\tilde{x}_{t_i \to t_{i+h}} &\approx \tilde{x}_{t_i} + \sum_{n=i-(k-h+1)}^{i-1} h(\epsilon_\theta(\tilde{x}_{t_n}, t_n), \tilde{x}_{t_n}, t_n) \cdot (t_{n+1} - t_n) \\
&= \phi\left(\{\epsilon_\theta(\tilde{x}_{\hat{t}_n}, \hat{t}_n)\}_{n=0}^{p-1}, t_{i-(k-h+1)}, t_i\right), \tilde{x}_{t_i}, t_i, t_{i+h}),
\end{aligned} \tag{B.9}$$

where $\phi$ is any $p$-order ODE solver. Eq. (B.9) demonstrates that under Assumption B.2, PFDiff-$k\_h$ utilizes past scores to replace current scores, converging to the same data distribution as that of any $p$-order ODE solver $\phi$. However, as noted in Remark 2, the time step size $\Delta t$ is very large (NFE<10), making "springboard" $\tilde{x}_{t_{i+h}}$ unreliable in Eq. (B.9). Therefore, we only use $\tilde{x}_{t_{i+h}}$ to predict a *foresight* update direction (i.e., the future score). The introduction of the future score can reduce the impact of neglecting higher-order derivative terms, thus correcting the discretization errors of the first-order ODE solvers.

**Convergence of PFDiff-$k\_h$ using future scores.** As described in Sec. 3.3, higher-order ODE solvers and future scores reduce the discretization error caused by neglecting higher-order derivative terms in two parallel manners. Therefore, PFDiff combines a higher-order ODE solver using only past scores, with convergence guarantees based on Assumption B.2. Next, we consider an iteration process of a first-order ODE solver from $\tilde{x}_{t_i}$ to $\tilde{x}_{t_{i+(k+1)}}$, which can be expressed as:

$$\begin{aligned}
\tilde{x}_{t_{i+(k+1)}} &= \tilde{x}_{t_i} + \int_{t_i}^{t_{i+(k+1)}} s(\epsilon_\theta(x_t,t), x_t, t)\mathrm{d}t \\
&\approx \tilde{x}_{t_i} + h(\epsilon_\theta(\tilde{x}_{t_i}, t_i), \tilde{x}_{t_i}, t_i) \cdot (t_{i+(k+1)} - t_i) \\
&= \phi(\epsilon_\theta(\tilde{x}_{t_i}, t_i), \tilde{x}_{t_i}, t_i, t_{i+(k+1)}),
\end{aligned} \tag{B.10}$$

where the second line is obtained by discretizing the first line with an existing first-order ODE solver, and the definitions of $\phi$ and $h$ are consistent with Sec. 3.1. It is well known that the "$\approx$" in Eq. (B.10) introduces discretization errors. We have revised Eq. (B.10) based on Assumption B.3, as follows:

$$\begin{aligned}
\tilde{x}_{t_{i+(k+1)}} &= \tilde{x}_{t_i} + \int_{t_i}^{t_{i+(k+1)}} s(\epsilon_\theta(x_t,t), x_t, t)\mathrm{d}t \\
&= \tilde{x}_{t_i} + s(\epsilon_\theta(\tilde{x}_{t_{\tilde{s}}}, t_{\tilde{s}}), \tilde{x}_{t_{\tilde{s}}}, t_{\tilde{s}}) \cdot (t_{i+(k+1)} - t_i) \\
&= \tilde{x}_{t_i} + h(\epsilon_\theta(\tilde{x}_{t_s}, t_s), \tilde{x}_{t_s}, t_s) \cdot (t_{i+(k+1)} - t_i) \\
&= \phi(\epsilon_\theta(\tilde{x}_{t_s}, t_s), \tilde{x}_{t_i}, t_i, t_{i+(k+1)}).
\end{aligned} \tag{B.11}$$

Eq. (B.11) indicates that there is an optimal time point $t_s \in [t_i, t_{i+(k+1)}]$ corresponding to the optimal score $\epsilon_\theta(\tilde{x}_{t_s}, t_s)$ that can correct the discretization error of Eq. (B.10). Furthermore, when the time step size $\Delta t = t_{i+(k+1)} - t_i$ is very large (for example, NFE<10), using the score at the current time point $t_i$ leads to non-convergence of the sampling process. This implies that the sampling trajectory of DPMs is not a straight line (if it were a straight line, a larger sampling step size could be used). Therefore, the optimal time point is not achieved at the endpoints, i.e., $t_s \neq t_i$ and $t_s \neq t_{i+(k+1)}$, and we adjust that $t_s$ falls within the interval $(t_i, t_{i+(k+1)})$. Additionally, to approximate the optimal score, we introduce the *foresight* update mechanism of the Nesterov momentum (Nesterov, 1983), and guide the current intermediate state sampling with future score information. In other words, we replace $\epsilon_\theta(\tilde{x}_{t_s}, t_s)$ with $\epsilon_\theta(\tilde{x}_{t_{i+h}}, t_{i+h})$, as follows:

$$
\begin{aligned}
\tilde{x}_{t_{i+(k+1)}} &= \phi(\epsilon_\theta(\tilde{x}_{t_s}, t_s), \tilde{x}_{t_i}, t_i, t_{i+(k+1)}) \\
&\approx \phi(\epsilon_\theta(\tilde{x}_{t_{i+h}}, t_{i+h}), \tilde{x}_{t_i}, t_i, t_{i+(k+1)}),
\end{aligned}
\tag{B.12}
$$

where $k$ and $h$ are determined by the selected PFDiff-$k$_$h$. According to the definition of PFDiff-$k$_$h$, $t_{i+h}$ also lies within the interval $(t_i, t_{i+(k+1)})$. For six different versions of PFDiff-$k$_$h$ defined in Appendix C, we believe the optimal $t_s$ within the interval $(t_i, t_{i+(k+1)})$ has been approximated, thereby completing the convergence proof of using future scores. Finally, we note that PFDiff using future scores to replace current scores is an approximation of the optimal score. Together with this section and Proposition 3.1 (future scores have less impact at neglecting higher-order derivative terms), we jointly verify that future scores can more effectively guide a first-order ODE solver in sampling.

## C  ALGORITHMS OF PFDIFFS

As described in Sec. 3.4, during a single iteration, we can leverage the *foresight* update mechanism to skip to a more distant future. Specifically, we modify Eq. (14) to $\tilde{x}_{t_{i+(k+1)}} \approx \phi(Q, \tilde{x}_{t_i}, t_i, t_{i+(k+1)})$ to achieve a $k$-step skip. We refer to this method as PFDiff-$k$. Additionally, when $k \neq 1$, the computation of the buffer $Q$, originating from Eq. (13), presents different selection choices. We modify Eq. (12) to $\tilde{x}_{t_{i+h}} \approx \phi(Q, \tilde{x}_{t_i}, t_i, t_{i+h}), h \leq k$ to denote different "*springboard*"

---

**Algorithm 2** PFDiff-2

**Require:** initial value $x_T$, NFE $N$, model $\epsilon_\theta$, any $p$-order solver $\phi$, skip type $h$
1: Define time steps $\{t_i\}_{i=0}^M$ with $M = 3N - 2p$
2: $\tilde{x}_{t_0} \leftarrow x_T$
3: $Q \xleftarrow{\text{buffer}} \left( \{\epsilon_\theta(\tilde{x}_{\hat{t}_n}, \hat{t}_n)\}_{n=0}^{p-1}, t_0, t_1 \right)$         ▷ Initialize buffer
4: $\tilde{x}_{t_1} = \phi(Q, \tilde{x}_{t_0}, t_0, t_1)$
5: **for** $i \leftarrow 1$ to $\frac{M}{p} - 3$ **do**
6:      **if** $(i-1) \bmod 3 = 0$ **then**
7:          **if** $h = 1$ **then**                                              ▷ PFDiff-2_1
8:             $\tilde{x}_{t_{i+1}} = \phi(Q, \tilde{x}_{t_i}, t_i, t_{i+1})$         ▷ Updating guided by past scores
9:             $Q \xleftarrow{\text{buffer}} \left( \{\epsilon_\theta(\tilde{x}_{\hat{t}_n}, \hat{t}_n)\}_{n=0}^{p-1}, t_{i+1}, t_{i+3} \right)$      ▷ Update buffer (overwrite)
10:          **else if** $h = 2$ **then**                                     ▷ PFDiff-2_2
11:             $\tilde{x}_{t_{i+2}} = \phi(Q, \tilde{x}_{t_i}, t_i, t_{i+2})$         ▷ Updating guided by past scores
12:             $Q \xleftarrow{\text{buffer}} \left( \{\epsilon_\theta(\tilde{x}_{\hat{t}_n}, \hat{t}_n)\}_{n=0}^{p-1}, t_{i+2}, t_{i+3} \right)$      ▷ Update buffer (overwrite)
13:          **end if**
14:          **if** $p = 1$ **then**
15:             $\tilde{x}_{t_{i+3}} = \phi(Q, \tilde{x}_{t_i}, t_i, t_{i+3})$       ▷ Anticipatory updating guided by future scores
16:          **else if** $p > 1$ **then**
17:             $\tilde{x}_{t_{i+3}} = \phi(Q, \tilde{x}_{t_{i+h}}, t_{i+h}, t_{i+3})$      ▷ The higher-order solver uses only past scores
18:          **end if**
19:      **end if**
20: **end for**
21: **return** $\tilde{x}_{t_M}$

---

---

**Algorithm 3** PFDiff-3

---

**Require:** initial value $x_T$, NFE $N$, model $\epsilon_\theta$, any $p$-order solver $\phi$, skip type $h$
1:  Define time steps $\{t_i\}_{i=0}^M$ with $M = 4N - 3p$
2:  $\tilde{x}_{t_0} \leftarrow x_T$
3:  $Q \xleftarrow{\text{buffer}} \left( \{ \epsilon_\theta(\tilde{x}_{\hat{t}_n}, \hat{t}_n) \}_{n=0}^{p-1}, t_0, t_1 \right)$            ▷ Initialize buffer
4:  $\tilde{x}_{t_1} = \phi(Q, \tilde{x}_{t_0}, t_0, t_1)$
5:  **for** $i \leftarrow 1$ to $\frac{M}{p} - 4$ **do**
6:       **if** $(i-1) \mod 4 = 0$ **then**
7:           $\tilde{x}_{t_{i+4}} = \phi(Q, \tilde{x}_{t_i}, t_i, t_{i+h})$          ▷ Updating guided by past scores
8:           $Q \xleftarrow{\text{buffer}} \left( \{ \epsilon_\theta(\tilde{x}_{\hat{t}_n}, \hat{t}_n) \}_{n=0}^{p-1}, t_{i+h}, t_{i+4} \right)$      ▷ Update buffer (overwrite)
9:           **if** $p = 1$ **then**
10:             $\tilde{x}_{t_{i+4}} = \phi(Q, \tilde{x}_{t_i}, t_i, t_{i+4})$     ▷ Anticipatory updating guided by future scores
11:          **else if** $p > 1$ **then**
12:             $\tilde{x}_{t_{i+4}} = \phi(Q, \tilde{x}_{t_{i+h}}, t_{i+h}, t_{i+4})$    ▷ The higher-order solver uses only past scores
13:          **end if**
14:      **end if**
15: **end for**
16: **return** $\tilde{x}_{t_M}$

---

choices with the parameter $h$. This strategy of multi-step skips and varying "springboard" choices is collectively termed as PFDiff-$k$_$h$ ($h \leq k$). Consequently, based on modifications to parameters $k$ and $h$ in Eq. (12) and Eq. (14), Eq. (13) is updated to $Q \xleftarrow{\text{buffer}} \left( \{ \epsilon_\theta(\tilde{x}_{\hat{t}_n}, \hat{t}_n) \}_{n=0}^{p-1}, t_{i+h}, t_{i+(k+1)} \right)$, and Eq. (11) is updated to $Q \xleftarrow{\text{buffer}} \left( \{ \epsilon_\theta(\tilde{x}_{\hat{t}_n}, \hat{t}_n) \}_{n=0}^{p-1}, t_{i-(k-h+1)}, t_i \right)$.

When $k = 1$, since $h \leq k$, then $h = 1$, and PFDiff-$k$_$h$ is the same as PFDiff-1, as shown in Algorithm 1 in Sec. 3.4. When $k = 2$, $h$ can be either 1 or 2, forming Algorithms PFDiff-2_1 and PFDiff-2_2, as shown in Algorithm 2. Furthermore, when $k = 3$, this forms three different versions of PFDiff-3, as shown in Algorithm 3. In this study, we utilize the optimal results from the six configurations of PFDiff-$k$_$h$ ($k = 1, 2, 3$ ($h \leq k$)) to demonstrate the performance of PFDiff. As described in Appendix B.3, this is essentially an approximation of the optimal time point $t_s$. Through these six different algorithm configurations, we approximately search for the optimal $t_s$. It is important to note that despite using six different algorithm configurations, this does not increase the computational burden in practical applications. This is because, by visual analysis of a small number of generated images or computing specific evaluation metrics, one can effectively select the algorithm configuration with the best performance. Moreover, even without any selection, with $k = 2$ and $h = 1$ fixed, PFDiff-2_1 can always improve the baseline's sampling quality within the range of 4~20 NFEs, as shown in the ablation study results in Sec. 4.3.

## D  ADDITIONAL EXPERIMENT RESULTS

In this section, we provide further supplements to the experiments on both unconditional and conditional pre-trained Diffusion Probabilistic Models (DPMs) as mentioned in Sec. 4. Through these additional supplementary experiments, we more fully validate the effectiveness of PFDiff as an orthogonal and training-free sampler. Unless otherwise stated, the selection of pre-trained DPMs, choice of baselines, algorithm configurations, GPU utilization, and other related aspects in this section are consistent with those described in Sec. 4.

### D.1  LICENSE

In this section, we list the used datasets, codes, and their licenses in Table 2.

Table 2: The used datasets, codes, and their licenses.

| Name | URL | License |
|---|---|---|
| CIFAR10 (Krizhevsky et al., 2009) | cs.toronto.edu | \ |
| CelebA 64x64 (Liu et al., 2015) | mmlab.ie.cuhk.edu.hk | \ |
| LSUN-Bedroom (Yu et al., 2015) | yf.io | \ |
| LSUN-Church (Yu et al., 2015) | yf.io | \ |
| ImageNet (Deng et al., 2009) | image-net.org | \ |
| MS-COCO2014 (Lin et al., 2014) | cocodataset.org | CC BY 4.0 |
| ScoreSDE (Song et al., 2020b) | github.com/yang-song | Apache-2.0 |
| DDIM (Song et al., 2020a) | github.com/ermongroup | MIT |
| Analytic-DPM (Bao et al., 2022b) | github.com/baofff | \ |
| DPM-Solver (Lu et al., 2022a) | github.com/LuChengTHU | MIT |
| DPM-Solver++ (Lu et al., 2022b) | github.com/LuChengTHU | MIT |
| Guided-Diffusion (Dhariwal & Nichol, 2021) | github.com/openai | MIT |
| Stable-Diffusion (Rombach et al., 2022) | github.com/CompVis | CreativeML Open RAIL-M |

Table 3: Sample quality measured by FID↓ on the CIFAR10 (Krizhevsky et al., 2009), CelebA 64x64 (Liu et al., 2015), LSUN-bedroom 256x256 (Yu et al., 2015), and LSUN-church 256x256 (Yu et al., 2015) datasets using unconditional discrete-time DPMs, varying the number of function evaluations (NFE). Evaluated on 50k samples. PFDiff uses DDIM (Song et al., 2020a) and Analytic-DDIM (Bao et al., 2022b) as baselines and introduces DDPM (Ho et al., 2020) and Analytic-DDPM (Bao et al., 2022b) with $\eta = 1.0$ from Eq. (6) for comparison.

| +PFDiff | Method | NFE | | | | | | |
|---|---|---|---|---|---|---|---|---|
| | | 4 | 6 | 8 | 10 | 12 | 15 | 20 |
| CIFAR10 (discrete-time model (Ho et al., 2020), quadratic time steps) | | | | | | | | |
| × | DDPM($\eta = 1.0$) (Ho et al., 2020) | 108.05 | 71.47 | 52.87 | 41.18 | 32.98 | 25.59 | 18.34 |
| × | Analytic-DDPM (Bao et al., 2022b) | 65.81 | 56.37 | 44.09 | 34.95 | 29.96 | 23.26 | 17.32 |
| × | Analytic-DDIM (Bao et al., 2022b) | 106.86 | 24.02 | 14.21 | 10.09 | 8.80 | 7.25 | 6.17 |
| × | DDIM (Song et al., 2020a) | 65.70 | 29.68 | 18.45 | 13.66 | 11.01 | 8.80 | 7.04 |
| ✓ | Analytic-DDIM | 289.84 | 23.24 | 7.03 | **4.51** | **3.91** | **3.75** | **3.65** |
| ✓ | DDIM | **22.38** | **9.84** | **5.64** | 4.57 | 4.39 | 4.10 | 3.68 |
| CelebA 64x64 (discrete-time model (Song et al., 2020a), quadratic time steps) | | | | | | | | |
| × | DDPM($\eta = 1.0$) (Ho et al., 2020) | 59.38 | 43.63 | 34.12 | 28.21 | 24.40 | 20.19 | 15.85 |
| × | Analytic-DDPM (Bao et al., 2022b) | 32.10 | 39.78 | 32.29 | 26.96 | 23.03 | 19.36 | 15.67 |
| × | Analytic-DDIM (Bao et al., 2022b) | 69.75 | 16.60 | 11.84 | 9.37 | 7.95 | 6.92 | 5.84 |
| × | DDIM (Song et al., 2020a) | 37.76 | 20.99 | 14.10 | 10.86 | 9.01 | 7.67 | 6.50 |
| ✓ | Analytic-DDIM | 360.21 | 28.24 | 5.66 | 4.90 | 4.62 | **4.55** | **4.55** |
| ✓ | DDIM | **13.29** | **7.53** | **5.06** | **4.71** | **4.60** | 4.70 | 4.68 |
| CelebA 64x64 (discrete-time model (Song et al., 2020a), uniform time steps) | | | | | | | | |
| × | DDPM($\eta = 1.0$) (Ho et al., 2020) | 65.39 | 49.52 | 41.65 | 36.68 | 33.45 | 30.27 | 26.76 |
| × | Analytic-DDPM (Bao et al., 2022b) | 102.45 | 42.43 | 34.36 | 33.85 | 30.38 | 28.90 | 25.89 |
| × | Analytic-DDIM (Bao et al., 2022b) | 90.44 | 24.85 | 16.45 | 16.67 | 15.11 | 15.00 | 13.40 |
| × | DDIM (Song et al., 2020a) | **44.36** | 29.12 | 23.19 | 20.50 | 18.43 | 16.71 | 14.76 |
| ✓ | Analytic-DDIM | 308.58 | 56.04 | 14.07 | 10.98 | 8.97 | **6.39** | **5.19** |
| ✓ | DDIM | 51.87 | **12.79** | **8.82** | **8.93** | **7.70** | 6.44 | 5.66 |
| LSUN-bedroom 256x256 (discrete-time model (Ho et al., 2020), uniform time steps) | | | | | | | | |
| × | DDIM (Song et al., 2020a) | **115.63** | 47.40 | 26.73 | 19.26 | 15.23 | 11.68 | 9.26 |
| ✓ | DDIM | 140.40 | **18.72** | **11.50** | **9.28** | **8.36** | **7.76** | **7.14** |
| LSUN-church 256x256 (discrete-time model (Ho et al., 2020), uniform time steps) | | | | | | | | |
| × | DDIM (Song et al., 2020a) | 121.95 | 50.02 | 30.04 | 22.04 | 17.66 | 14.58 | 12.49 |
| ✓ | DDIM | **72.86** | **18.30** | **14.34** | **13.27** | **12.05** | **11.77** | **11.12** |

## D.2 ADDITIONAL RESULTS FOR UNCONDITIONAL DISCRETE-TIME SAMPLING

In this section, we report on experiments with unconditional, discrete DPMs on the CI-FAR10 (Krizhevsky et al., 2009) and CelebA 64x64 (Liu et al., 2015) datasets using quadratic time steps. The FID↓ scores for the PFDiff algorithm are reported for changes in the number of function evaluations (NFE) from 4 to 20. Additionally, we present FID scores on the CelebA 64x64 (Liu et al., 2015), LSUN-bedroom 256x256 (Yu et al., 2015), and LSUN-church 256x256 (Yu et al., 2015) datasets, utilizing uniform time steps. The experimental results are summarized in Table 3. Results indicate that using DDIM (Song et al., 2020a) as the baseline, our method (PFDiff) nearly achieved significant performance improvements across all datasets and NFE settings. Notably, PFDiff facilitates rapid convergence of pre-trained DPMs to the data distribution with NFE settings below 10, validating its effectiveness on discrete pre-trained DPMs and the first-order ODE solver DDIM. It is important to note that on the CIFAR10 and CelebA 64x64 datasets, we have included the FID scores of Analytic-DDIM (Bao et al., 2022b), which serves as another baseline. Analytic-DDIM modifies the variance in DDIM and introduces some random noise. With NFE lower than 10, the presence of minimal random noise amplifies the error introduced by the score information approximation in PFDiff, reducing its error correction capability compared to the Analytic-DDIM sampler. Thus, in fewer-step sampling (NFE<10), using DDIM as the baseline is more effective than using Analytic-DDIM, which requires recalculating the optimal variance for different pre-trained DPMs, thereby introducing additional computational overhead. In other experiments with pre-trained DPMs, we validate the efficacy of the PFDiff algorithm by combining it with the overall superior performance of the DDIM solver.

Furthermore, to validate the motivation proposed in Sec. 3.2 based on Fig. 1a—that at not excessively large time step size $\Delta t$, an ODE-based solver shows considerable similarity in the noise network outputs—we compare it with the SDE-based solver DDPM (Ho et al., 2020). Even at smaller $\Delta t$, the mean squared error (MSE) of the noise outputs from DDPM remains high, suggesting that the effectiveness of PFDiff may be limited when based on SDE solvers. Further, we adjusted the $\eta$ parameter in Eq. (6) (which controls the amount of noise introduced in DDPM) from 1.0 to 0.0 (at

Table 4: Sample quality measured by FID↓ on the CIFAR10 (Krizhevsky et al., 2009) and CelebA 64x64 (Liu et al., 2015) using unconditional discrete-time DPMs with and without our method (PFDiff), varying the number of function evaluations (NFE) and $\eta$ from Eq. (6). Evaluated on 50k samples.

| Method | NFE | | | | | | |
|---|---|---|---|---|---|---|---|
| | 4 | 6 | 8 | 10 | 12 | 15 | 20 |
| CIFAR10 (discrete-time model (Ho et al., 2020), quadratic time steps) | | | | | | | |
| DDPM($\eta = 1.0$) (Ho et al., 2020) | 108.05 | 71.47 | 52.87 | 41.18 | 32.98 | 25.59 | 18.34 |
| +PFDiff (Ours) | 475.47 | 432.24 | 344.96 | 332.41 | 285.88 | 158.90 | 28.05 |
| DDPM($\eta = 0.5$) (Song et al., 2020a) | 71.08 | 34.32 | 22.37 | 16.63 | 13.37 | 10.75 | 8.38 |
| +PFDiff (Ours) | 432.50 | 349.09 | 311.62 | 167.65 | 59.93 | 23.17 | 10.61 |
| DDPM($\eta = 0.2$) (Song et al., 2020a) | 66.33 | 30.26 | 18.94 | 14.01 | 11.25 | 9.00 | 7.18 |
| +PFDiff (Ours) | 316.15 | 189.02 | 18.55 | 7.73 | 5.70 | 4.53 | 4.00 |
| DDIM($\eta = 0.0$) (Song et al., 2020a) | 65.70 | 29.68 | 18.45 | 13.66 | 11.01 | 8.80 | 7.04 |
| +PFDiff (Ours) | **22.38** | **9.48** | **5.64** | **4.57** | **4.39** | **4.10** | **3.68** |
| CelebA 64x64 (discrete-time model (Song et al., 2020a), quadratic time steps) | | | | | | | |
| DDPM($\eta = 1.0$) (Ho et al., 2020) | 59.38 | 43.63 | 34.12 | 28.21 | 24.40 | 20.19 | 15.85 |
| +PFDiff (Ours) | 433.25 | 439.19 | 415.41 | 317.43 | 324.58 | 326.50 | 171.41 |
| DDPM($\eta = 0.5$) (Song et al., 2020a) | 40.58 | 23.72 | 16.74 | 13.15 | 11.27 | 9.36 | 7.73 |
| +PFDiff (Ours) | 435.27 | 417.58 | 314.63 | 310.10 | 252.19 | 69.31 | 19.23 |
| DDPM($\eta = 0.2$) (Song et al., 2020a) | 38.20 | 21.35 | 14.55 | 11.22 | 9.47 | 7.99 | 6.71 |
| +PFDiff (Ours) | 394.03 | 319.02 | 45.15 | 12.71 | 7.85 | 5.10 | 4.96 |
| DDIM($\eta = 0.0$) (Song et al., 2020a) | 37.76 | 20.99 | 14.10 | 10.86 | 9.01 | 7.67 | 6.50 |
| +PFDiff (Ours) | **13.29** | **7.53** | **5.06** | **4.71** | **4.60** | **4.70** | **4.68** |

$\eta = 0.0$, the SDE-based DDPM degenerates into the ODE-based DDIM (Song et al., 2020a)). As shown in Fig. 1a, as $\eta$ decreases, the MSE of the noise network outputs gradually decreases at the same time step size $\Delta t$, indicating that reducing noise introduction can enhance the effectiveness of PFDiff. To verify this motivation, we utilized quadratic time steps on CIFAR10 and CelebA 64x64 datasets and controlled the amount of noise introduced by adjusting $\eta$, to demonstrate that PFDiff can leverage the temporal redundancy present in ODE solvers to boost its performance. The experimental results, as shown in Table 4, illustrate that with the reduction of $\eta$ from 1.0 (SDE) to 0.0 (ODE), PFDiff's sampling performance significantly improves at fewer time steps (NFE≤20). The experiment results regarding FID variations with NFE as presented in Table 4, align with the trends of MSE of noise network outputs with changes in time step size $\Delta t$ as depicted in Fig. 1a. This reaffirms the motivation we proposed in Sec. 3.2.

### D.3 ADDITIONAL RESULTS FOR UNCONDITIONAL CONTINUOUS-TIME SAMPLING

In this section, we supplement the specific FID↓ scores for the unconditional, continuous pre-trained DPMs models with first-order and higher-order ODE solvers, DPM-Solver-1, -2 and -3, (Lu et al., 2022a) as baselines, as shown in Table 5. For all experiments in this section, we conducted tests on the CIFAR10 dataset (Krizhevsky et al., 2009), using the checkpoint `checkpoint_8.pth` under the `vp/cifar10_ddpmpp_deep_continuous` configuration provided by ScoreSDE (Song et al., 2020b). For the hyperparameter `method` of DPM-Solver (Lu et al., 2022a), we adopted `singlestep_fixed`; to maintain consistency with the discrete-time model in Appendix D.2, the parameter `skip` was set to `time_quadratic` (i.e., quadratic time steps). Unless otherwise specified, we used the parameter settings recommended by DPM-Solver. The results in Table 5 show that by using the PFDiff method described in Sec. 3.4 and taking DPM-Solver as the baseline, we were able to further enhance sampling performance on the basis of first-order and higher-order ODE solvers. Particularly, in the 6∼12 NFE range, PFDiff significantly improved the convergence issues of higher-order ODE solvers under fewer NFEs. For instance, at 9 NFE, PFDiff reduced the FID of DPM-Solver-3 from 233.56 to 5.67, improving the sampling quality by 97.57%. These results validate the effectiveness of using PFDiff with first-order or higher-order ODE solvers as the baseline.

Table 5: Sample quality measured by FID↓ of different orders of DPM-Solver (Lu et al., 2022a) on the CIFAR10 (Krizhevsky et al., 2009) using unconditional continuous-time DPMs with and without our method (PFDiff), varying the number of function evaluations (NFE). Evaluated on 50k samples.

| Method | order | NFE | | | | | | |
|---|---|---|---|---|---|---|---|---|
| | | 4 | 6 | 8 | 10 | 12 | 16 | 20 |
| CIFAR10 (continuous-time model (Song et al., 2020b), quadratic time steps) | | | | | | | | |
| DPM-Solver-1 (Lu et al., 2022a) | 1 | **40.55** | 23.86 | 15.57 | 11.64 | 9.64 | 7.23 | 6.06 |
| +PFDiff (Ours) | 1 | 113.74 | **11.41** | **5.90** | **4.23** | **3.92** | **3.73** | **3.75** |
| DPM-Solver-2 (Lu et al., 2022a) | 2 | 298.79 | 106.05 | 41.79 | 14.43 | 6.75 | **4.24** | **3.91** |
| +PFDiff (Ours) | 2 | **85.22** | **16.30** | **9.67** | **6.64** | **5.74** | 5.12 | 4.78 |
| | | | 6 | 9 | | 12 | 15 | 21 |
| DPM-Solver-3 (Lu et al., 2022a) | 3 | | 382.51 | 233.56 | | 44.82 | 7.98 | **3.63** |
| +PFDiff (Ours) | 3 | | **103.22** | **5.67** | | **5.72** | **5.62** | 5.24 |

### D.4 ADDITIONAL RESULTS FOR CLASSIFIER GUIDANCE

In this section, we provide the specific FID scores for pre-trained DPMs in the conditional, classifier guidance paradigm on the ImageNet 64x64 (Deng et al., 2009) and ImageNet 256x256 datasets (Deng et al., 2009), as shown in Table 6. We now describe the experimental setup in detail. For the pre-trained models, we used the ADM-G (Dhariwal & Nichol, 2021) provided `64x64_diffusion.pt` and `64x64_classifier.pt` for the ImageNet 64x64 dataset, and `256x256_diffusion.pt` and `256x256_classifier.pt` for the ImageNet 256x256

Table 6: Sample quality measured by FID↓ on the ImageNet 64x64 (Deng et al., 2009) and ImageNet 256x256 (Deng et al., 2009), using ADM-G (Dhariwal & Nichol, 2021) model with guidance scales ($s$) of 1.0 and 2.0, varying the number of function evaluations (NFE). Evaluated: ImageNet 64x64 with 50k, ImageNet 256x256 with 10k samples. *We directly borrowed the results reported by AutoDiffusion (Li et al., 2023), and AutoDiffusion requires additional search costs. "\" represents missing data in the original paper.

| Method | Step | NFE | | | | | |
|---|---|---|---|---|---|---|---|
| | | 4 | 6 | 8 | 10 | 15 | 20 |
| ImageNet 64x64 (pixel DPMs model (Dhariwal & Nichol, 2021), uniform time steps, $s = 1.0$) | | | | | | | |
| DDIM (Song et al., 2020a) | S | 138.81 | 23.58 | 12.54 | 8.93 | 5.52 | 4.45 |
| DPM-Solver-2 (Lu et al., 2022a) | S | 327.09 | 292.66 | 264.97 | 236.80 | 166.52 | 120.29 |
| DPM-Solver-2 (Lu et al., 2022a) | M | 48.64 | 21.08 | 12.45 | 8.86 | 5.57 | 4.46 |
| DPM-Solver-3 (Lu et al., 2022a) | S | 383.71 | 376.86 | 380.51 | 378.32 | 339.34 | 280.12 |
| DPM-Solver-3 (Lu et al., 2022a) | M | 54.01 | 24.76 | 13.17 | 8.85 | 5.48 | 4.41 |
| DPM-Solver++(2M) (Lu et al., 2022b) | M | 44.15 | 20.44 | 12.53 | 8.95 | 5.53 | 4.33 |
| *AutoDiffusion (Li et al., 2023) | S | 17.86 | 11.17 | \ | 6.24 | 4.92 | 3.93 |
| DDIM+PFDiff (Ours) | S | **16.46** | **8.20** | **6.22** | **5.19** | **4.20** | **3.83** |
| ImageNet 256x256 (pixel DPMs model (Dhariwal & Nichol, 2021), uniform time steps, $s = 2.0$) | | | | | | | |
| DDIM (Song et al., 2020a) | S | 51.79 | 23.48 | 16.33 | 12.93 | 9.89 | 9.05 |
| DDIM+PFDiff (Ours) | S | **37.81** | **18.15** | **12.22** | **10.33** | **8.59** | **8.08** |

dataset. All experiments were conducted with uniform time steps and used DDIM as the baseline (Song et al., 2020a). We implemented the second-order and third-order methods from DPM-Solver (Lu et al., 2022a) for comparison and explored the `method` hyperparameter provided by DPM-Solver for both `singlestep` (corresponding to "S" in Table 6) and `multistep` (corresponding to "M" in Table 6). Additionally, we implemented the best-performing method from DPM-Solver++ (Lu et al., 2022b), multi-step DPM-Solver++(2M), as a comparative measure. Furthermore, we also introduced the superior-performing AutoDiffusion (Li et al., 2023) method as a comparison. *We directly borrowed the results reported in the original paper, emphasizing that although AutoDiffusion does not require additional training, it incurs additional search costs. "\" represents missing data in the original paper. The specific experimental results of the configurations mentioned are shown in Table 6. The results demonstrate that PFDiff, using DDIM as the baseline on the ImageNet 64x64 dataset, significantly enhances the sampling efficiency of DDIM and surpasses previous optimal training-free sampling methods. Particularly, in cases where NFE≤10, PFDiff improved the sampling quality of DDIM by 41.88%∼88.14%. Moreover, on the large ImageNet 256x256 dataset, PFDiff demonstrates a consistent performance improvement over the DDIM baseline, similar to the improvements observed on the ImageNet 64x64 dataset.

## D.5 ADDITIONAL RESULTS FOR CLASSIFIER-FREE GUIDANCE

In this section, we supplemented the specific FID↓ scores for the Stable-Diffusion (Rombach et al., 2022) (conditional, classifier-free guidance paradigm) setting with a guidance scale ($s$) of 7.5 and 1.5. Specifically, for the pre-trained model, we conducted experiments using the `sd-v1-4.ckpt` checkpoint provided by Stable-Diffusion. All experiments used the MS-COCO2014 (Lin et al., 2014) validation set to calculate FID↓ scores, with uniform time steps. PFDiff employs the DDIM (Song et al., 2020a) method as the baseline. Initially, under the recommended $s = 7.5$ configuration by Stable-Diffusion, we implemented DPM-Solver-2 and -3 as comparative methods, and set the `method` hyperparameters provided by DPM-Solver to `multistep` (corresponding to "M" in Table 7). Additionally, we introduced previous state-of-the-art training-free methods, including DPM-Solver++(2M) (Lu et al., 2022b), UniPC (Zhao et al., 2023), and DPM-Solver-v3(2M) (Zheng et al., 2023) for comparison. The experimental results are shown in Table 7. †We borrow the results reported in DPM-Solver-v3 (Zheng et al., 2023) directly. The results indicate that on Stable-Diffusion, PFDiff, using only DDIM as a baseline, surpasses the previous state-of-the-art training-free sampling methods in terms of sampling quality in fewer steps (NFE<20). Particularly, at NFE=10, PFDiff achieved a 13.06 FID, nearly converging to the data distribution, which is a

Table 7: Sample quality measured by FID↓ on the validation set of MS-COCO2014 (Lin et al., 2014) using Stable-Diffusion model (Rombach et al., 2022) with guidance scales ($s$) of 7.5 and 1.5, varying the number of function evaluations (NFE). Evaluated on 10k samples. [†]We borrow the results reported in DPM-Solver-v3 (Zheng et al., 2023) directly.

| Method | Step | NFE | | | | | |
|---|---|---|---|---|---|---|---|
| | | 5 | 6 | 8 | 10 | 15 | 20 |
| MS-COCO2014 (latent DPMs model (Rombach et al., 2022), uniform time steps, $s = 7.5$) | | | | | | | |
| DDIM (Song et al., 2020a) | S | 23.92 | 20.33 | 17.46 | 16.78 | 16.08 | 15.95 |
| DPM-Solver-2 (Lu et al., 2022a) | M | 18.97 | 17.37 | 16.29 | 15.99 | 14.32 | 14.38 |
| DPM-Solver-3 (Lu et al., 2022a) | M | 18.89 | 17.34 | 16.25 | 16.11 | 14.10 | **13.44** |
| [†]DPM-Solver++(2M) (Lu et al., 2022b) | M | 18.87 | 17.44 | 16.40 | 15.93 | 15.84 | 15.72 |
| [†]UniPC (Zhao et al., 2023) | M | 18.77 | 17.32 | 16.20 | 16.15 | 16.06 | 15.94 |
| [†]DPM-Solver-v3(2M) (Zheng et al., 2023) | M | 18.83 | 16.41 | 15.41 | 15.32 | 15.30 | 15.23 |
| DDIM+PFDiff (Ours) | S | **18.31** | **15.47** | **13.26** | **13.06** | **13.57** | 13.97 |
| MS-COCO2014 (latent DPMs model (Rombach et al., 2022), uniform time steps, $s = 1.5$) | | | | | | | |
| DDIM (Song et al., 2020a) | S | 70.36 | 54.32 | 37.54 | 29.41 | 20.54 | 18.17 |
| DPM-Solver-2 (Lu et al., 2022a) | M | 37.47 | 27.79 | 19.65 | 18.39 | 17.27 | 16.85 |
| DPM-Solver-3 (Lu et al., 2022a) | M | 35.90 | 25.88 | 18.26 | 19.10 | 17.21 | 16.67 |
| DPM-Solver++(2M) (Lu et al., 2022b) | M | 36.58 | 26.78 | 18.92 | 20.26 | 18.61 | 17.78 |
| DDIM+PFDiff (Ours) | S | **24.31** | **20.99** | **18.09** | **17.00** | **16.03** | **15.57** |

14.25% improvement over the previous state-of-the-art method DPM-Solver-v3 at 20 NFE, which had a 15.23 FID. Furthermore, to further validate the effectiveness of PFDiff on Stable-Diffusion, we conducted experiments using the $s = 1.5$ setting with the same experimental configuration as $s = 7.5$. For the comparative methods, we only experimented with the multi-step versions of DPM-Solver-2 and -3 and DPM-Solver++(2M), which had faster convergence at fewer NFE under the $s = 7.5$ setting. As for UniPC and DPM-Solver-v3(2M), since DPM-Solver-v3 did not provide corresponding experimental results at $s = 1.5$, we did not list their comparative results. The experimental results show that PFDiff, using DDIM as the baseline under the $s = 1.5$ setting, demonstrated consistent performance improvements as seen in the $s = 7.5$ setting, as shown in Table 7.

## D.6 ADDITIONAL ABLATION STUDY RESULTS

### D.6.1 ADDITIONAL RESULTS FOR PFDIFF HYPERPARAMETERS STUDY

In this section, we extensively investigate the impact of the hyperparameters $k$ and $h$ on the performance of the PFDiff algorithm, supplementing with the results of ablation experiments and experimental setups. Specifically, for the unconditional DPMs, we conducted experiments on the CIFAR10 dataset (Krizhevsky et al., 2009) using quadratic time steps, based on pre-trained unconditional discrete DDPM (Ho et al., 2020). For the conditional DPMs, we used uniform time steps in classifier guidance ADM-G (Dhariwal & Nichol, 2021) pre-trained DPMs, setting the guidance scale ($s$) to 1.0 for experiments on the ImageNet 64x64 dataset (Deng et al., 2009). All experiments were conducted using the DDIM (Song et al., 2020a) algorithm as a baseline, and PFDiff-$k$_$h$ configurations ($k = 1, 2, 3$ ($h \leq k$)) were tested in six different algorithm configurations. The FID↓ scores are presented in Table 8, by varying the number of function evaluations (NFE) and the sample number used to compute the evaluation metrics.

We first analyze the impact of the hyperparameters $k$ and $h$ using 50k samples to compute the FID scores, which is a common method for evaluating the performance of sampling algorithms. The experimental results demonstrate that, under various combinations of $k$ and $h$, PFDiff is able to enhance the sampling performance of the DDIM baseline in most cases across different types of pre-trained DPMs. Particularly when setting $k = 2$ and $h = 1$, PFDiff-2_1 can always improve the sampling performance of the DDIM baseline within the range of 4~20 NFE. Furthermore, we have an exciting discovery regarding the further optimization of algorithm performance: Searching with just 1/10 of the data provides consistent results compared to searches using the full 50k samples,

Table 8: Ablation study of the impact of $k$ and $h$ on PFDiff in CIFAR10 (Krizhevsky et al., 2009) and ImageNet 64x64 (Deng et al., 2009) datasets using DDPM (Ho et al., 2020) and ADM-G (Dhariwal & Nichol, 2021) models. We report the FID↓ and MSE↓, varying the number of function evaluations (NFE) and the number of samples for evaluating algorithm performance.

| Samples | Method | NFE | | | | | | |
|---|---|---|---|---|---|---|---|---|
| | | 4 | 6 | 8 | 10 | 12 | 15 | 20 |
| CIFAR10 (unconditional DPMs model (Ho et al., 2020), quadratic time steps) | | | | | | | | |
| 50k (FID) | DDIM (Song et al., 2020a) | 65.70 | 29.68 | 18.45 | 13.66 | 11.01 | 8.80 | 7.04 |
| | +PFDiff-1 | 124.73 | 19.45 | 5.78 | 4.95 | 4.63 | 4.25 | 4.14 |
| | +PFDiff-2_1 | 59.61 | **9.84** | 7.01 | 6.31 | 5.58 | 5.18 | 4.78 |
| | +PFDiff-2_2 | 167.12 | 53.22 | 8.43 | 4.95 | 4.41 | **4.10** | 3.78 |
| | +PFDiff-3_1 | **22.38** | 13.40 | 9.40 | 7.70 | 6.73 | 6.03 | 5.05 |
| | +PFDiff-3_2 | 129.18 | 19.35 | **5.64** | **4.57** | **4.39** | 4.19 | 4.08 |
| | +PFDiff-3_3 | 205.87 | 76.62 | 20.84 | 5.71 | 4.73 | 4.41 | **3.68** |
| 5k (FID) | DDIM (Song et al., 2020a) | 69.79 | 34.20 | 22.84 | 17.39 | 15.56 | 12.87 | 11.62 |
| | +PFDiff-1 | 127.82 | 23.96 | 10.35 | 9.73 | 9.29 | 9.09 | 8.74 |
| | +PFDiff-2_1 | 63.59 | **14.34** | 11.40 | 11.08 | 9.23 | 10.03 | 9.38 |
| | +PFDiff-2_2 | 170.58 | 57.74 | 12.94 | 9.86 | 10.08 | **9.02** | 8.44 |
| | +PFDiff-3_1 | **26.85** | 17.93 | 13.95 | 12.37 | 9.59 | 10.89 | 9.65 |
| | +PFDiff-3_2 | 132.45 | 23.80 | **10.11** | **9.58** | **9.16** | 9.09 | 8.69 |
| | +PFDiff-3_3 | 208.80 | 80.13 | 24.84 | 10.73 | 11.16 | 9.14 | **8.34** |
| 256 (MSE) | DDIM (Song et al., 2020a) | 0.1009 | 0.0608 | 0.0414 | 0.0314 | 0.0255 | 0.0199 | 0.0152 |
| | +PFDiff-1 | 0.0542 | 0.0217 | 0.0131 | **0.0100** | **0.0089** | **0.0082** | **0.0081** |
| | +PFDiff-2_1 | 0.0277 | **0.0137** | **0.0110** | 0.0104 | 0.0098 | 0.0093 | 0.0088 |
| | +PFDiff-2_2 | 0.1001 | 0.0468 | 0.0277 | 0.0184 | 0.0145 | 0.0122 | 0.0105 |
| | +PFDiff-3_1 | **0.0218** | 0.0167 | 0.0146 | 0.0130 | 0.0120 | 0.0107 | 0.0098 |
| | +PFDiff-3_2 | 0.0614 | 0.0228 | 0.0133 | 0.0101 | **0.0089** | 0.0083 | 0.0082 |
| | +PFDiff-3_3 | 0.1790 | 0.0820 | 0.0444 | 0.0299 | 0.0224 | 0.0165 | 0.0126 |
| ImageNet 64x64 (conditional DPMs model (Dhariwal & Nichol, 2021), uniform time steps, $s = 1.0$) | | | | | | | | |
| 50k (FID) | DDIM (Song et al., 2020a) | 138.81 | 23.58 | 12.54 | 8.93 | 6.74 | 5.52 | 4.45 |
| | +PFDiff-1 | 26.86 | 11.39 | 7.47 | 5.83 | 5.16 | 4.76 | 4.39 |
| | +PFDiff-2_1 | 17.14 | 8.94 | 6.38 | 5.46 | 5.46 | 4.30 | **3.83** |
| | +PFDiff-2_2 | 23.66 | 9.93 | 6.86 | 5.72 | 5.17 | 4.49 | 3.94 |
| | +PFDiff-3_1 | 16.74 | 9.43 | 7.19 | 5.86 | 5.07 | 4.69 | 4.44 |
| | +PFDiff-3_2 | **16.46** | **8.20** | **6.22** | **5.19** | **4.62** | **4.20** | 4.28 |
| | +PFDiff-3_3 | 23.06 | 9.73 | 6.92 | 5.55 | 5.21 | 4.47 | 4.49 |
| 5k (FID) | DDIM (Song et al., 2020a) | 146.03 | 29.61 | 19.11 | 15.13 | 13.15 | 11.65 | 10.81 |
| | +PFDiff-1 | 32.82 | 17.80 | 13.61 | 12.16 | 11.20 | 10.99 | 10.82 |
| | +PFDiff-2_1 | 23.70 | 14.81 | 12.38 | 11.82 | 11.53 | 10.77 | **10.24** |
| | +PFDiff-2_2 | 30.10 | 16.35 | 13.09 | 11.80 | 11.68 | 10.67 | 10.56 |
| | +PFDiff-3_1 | 23.09 | 15.78 | 13.21 | 12.09 | 11.71 | 11.00 | 10.77 |
| | +PFDiff-3_2 | **22.54** | **14.23** | **12.24** | **11.27** | **11.16** | **10.47** | 10.48 |
| | +PFDiff-3_3 | 29.45 | 16.12 | 13.25 | 11.90 | 11.29 | 10.68 | 10.66 |

significantly reducing the cost of hyperparameter searching for $k$ and $h$. Specifically, for the same NFE, the optimal combinations of $k$ and $h$ based on FID scores are consistent for both 5k and 50k samples. For instance, when NFE=6, the best FID values for both 5k and 50k samples are achieved with $k = 2$ and $h = 1$. For the six combinations used in this study with $k \leq 3(h \leq k)$, only a total of 30k samples are required to search the optimal $k$ and $h$ combination for each NFE—this is even less than the cost of evaluating algorithm performance normally with 50k samples. Additionally, in practical applications where only a small number of samples are needed for visual analysis, we can minimize training resources and rapidly identify the optimal $k$ and $h$ combination. In summary, the hyperparameters $k$ and $h$ do not impede the practical application of PFDiff in accelerating the sampling of DPMs.

Furthermore, we propose an automatic search strategy with almost no additional cost based on truncation errors for selecting $k$ and $h$. Specifically, in Sec. 3.5, we discuss how PFDiff can correct

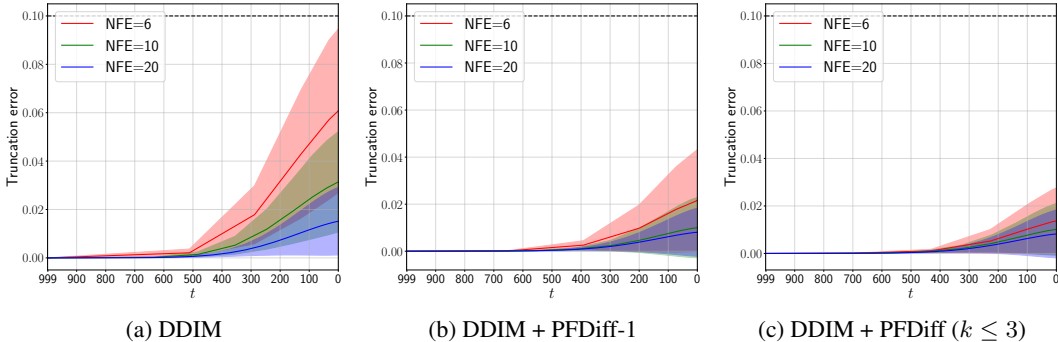

|  |  |  |
|---|---|---|
| (a) DDIM | (b) DDIM + PFDiff-1 | (c) DDIM + PFDiff ($k \leq 3$) |

Figure 5: The trend (mean and standard deviation (std)) of accumulated truncation error over time step $t$ on the CIFAR10 (Krizhevsky et al., 2009) dataset, relative to DDIM (Song et al., 2020a) with 1000 NFE, varying the number of function evaluations (NFE) $\in \{6, 10, 20\}$.

the discretization error of the baseline solver in regions with large curvature along the sampling trajectory, which means that PFDiff accumulates less truncation error. To validate this conclusion, we conducted experiments on the CIFAR10 dataset, where we sampled 256 samples using DDIM, PFDiff-1, and PFDiff ($k \leq 3$) with varying NFE values. The truncation errors were calculated using MSE relative to DDIM with 1000 NFE, and the results are shown in Fig. 5. The results demonstrate that PFDiff significantly corrects DDIM's truncation errors in regions with large curvature of the sampling trajectory (i.e., regions where truncation errors increase rapidly). Based on these findings, we propose an automatic search strategy: by varying the hyperparameters $k$ and $h$, we *"warm up" with only 256 samples* respectively, calculate the average truncation error using MSE, and use this value to guide the selection of $k$ and $h$. On the CIFAR10 dataset, by varying NFE, $k$, and $h$, we sampled 256 samples and computed the average truncation errors relative to DDIM with 1000 NFE. The results are presented in Tab. 8. Based on Tab. 8, we select the values of $k$ and $h$ corresponding to the minimum truncation error, and further refer to the corresponding FID values found in Tab. 8, which are $\{22.38, 9.84, 7.01, 4.95, 4.63/4.39, 4.25, 4.14\}$ for NFE $\in \{4, 6, 8, 10, 12, 15, 20\}$. This set of FID values is comparable to the optimal FID, indicating that determining $k$ and $h$ based on truncation error is reasonable. Notably, in Tab. 8, we "warmed up" only 256 samples to compute the truncation error, so the additional cost introduced by the automatic search strategy is negligible. The 256 samples are sufficient to capture the dataset's statistical properties because the shapes of sampling trajectories are highly similar (Chen et al., 2024).

### D.6.2  ABLATION STUDY OF PAST AND FUTURE SCORES

To further investigate the impact of scores from the past or future on first-order ODE solvers on the rapid updating of current intermediate states, this section supplements related ablation study results and their settings. Specifically, we first use PFDiff with the first-order ODE solver DDIM as a baseline, removing past and future scores separately based on the discrete-time pre-trained models (Ho et al., 2020; Song et al., 2020a). On the CIFAR10 (Krizhevsky et al., 2009) and CelebA 64x64 (Liu et al., 2015) datasets, we alter the number of function evaluations (NFE) to compute the FID↓ metric. Additionally, we introduce a method from previous literature (Ma et al., 2024) that accelerates sampling by caching part of past scores for comparison. Specifically, we configure the hyperparameters based on the DeepCache (Ma et al., 2024) codebase, setting `cache_interval` to 2 and `branch` to 0, with all other settings remaining unchanged. As shown in Table 9, the experimental results indicate that using only past scores or only future scores can slightly improve the first-order ODE solvers sampling performance. However, their combined use (i.e., the complete PFDiff) significantly enhances first-order ODE solvers sampling performance, especially with very few steps (NFE<10), a phenomenon particularly evident. These results further validate the efficiency of the PFDiff algorithm when NFE<10, benefiting from its information-efficient update process, which utilizes past and future (complementing each other) scores to jointly guide the current intermediate state.

Furthermore, to verify whether the update process of the PFDiff algorithm increases additional inference time, we employed the same experimental settings as in Table 9 and provided a specific

Table 9: Ablation study of the impact of the past and future scores on PFDiff, using DDIM (Song et al., 2020a) as the baseline, in CIFAR10 (Krizhevsky et al., 2009) and CelebA 64x64 (Liu et al., 2015) datasets using discrete-time models (Ho et al., 2020; Song et al., 2020a). We report the FID↓, varying the number of function evaluations (NFE). Evaluated on 50k samples.

| +PFDiff | Method | NFE | | | | | | |
|---|---|---|---|---|---|---|---|---|
| | | 4 | 6 | 8 | 10 | 12 | 16 | 20 |
| CIFAR10 (discrete-time model (Ho et al., 2020), quadratic time steps) | | | | | | | | |
| ✗ | DDIM (Song et al., 2020a) | 65.70 | 29.68 | 18.45 | 13.66 | 11.01 | 8.80 | 7.04 |
| ✗ | +Cache (Ma et al., 2024) | 49.02 | 24.04 | 15.23 | 11.31 | 9.40 | 7.25 | 6.25 |
| ✗ | +Past | 52.81 | 27.47 | 17.87 | 13.64 | 10.79 | 8.20 | 7.02 |
| ✗ | +Future | 66.06 | 25.39 | 11.93 | 8.06 | 6.04 | 4.17 | 4.07 |
| ✓ | +Past & Future | **22.38** | **9.84** | **5.64** | **4.57** | **4.39** | **4.10** | **3.68** |
| CelebA 64x64 (discrete-time model (Song et al., 2020a), uniform time steps) | | | | | | | | |
| ✗ | DDIM (Song et al., 2020a) | 44.36 | 29.12 | 23.19 | 20.50 | 18.43 | 16.13 | 14.76 |
| ✗ | +Cache (Ma et al., 2024) | 33.86 | 25.95 | 22.29 | 19.83 | 18.28 | 16.05 | 14.45 |
| ✗ | +Past | **28.45** | 21.56 | 18.65 | 17.03 | 15.89 | 14.05 | 12.73 |
| ✗ | +Future | 39.85 | 16.30 | 14.40 | 12.79 | 12.13 | 9.73 | 8.13 |
| ✓ | +Past & Future | 51.87 | **12.79** | **8.82** | **8.93** | **7.70** | **6.44** | **5.66** |

inference time comparison under different NFE. The inference time↓ (second, mean±std) required per 1k samples on a single NVIDIA 3090 GPU is shown in Table 10. The experimental results reveal that PFDiff+DDIM has consistent inference times with DDIM alone under the same NFE, indicating that the PFDiff algorithm does not add extra inference time. Additionally, methods (Ma et al., 2024) that cache part of past scores not only incur additional inference costs but also exhibit relatively weak acceleration effects with a small number of steps (NFE<10). These results collectively demonstrate that the PFDiff algorithm can significantly enhance sampling quality without any increase in inference time, further proving its effectiveness.

Table 10: Inference time↓ (second, mean±std) required per 1k samples on a single NVIDIA 3090 GPU, varying the number of function evaluations (NFE). We additionally present the inference time with only past or only future scores, at the same NFE. Moreover, we introduce methods (Ma et al., 2024; Wimbauer et al., 2024) that cache part of past scores for comparison.

| +PFDiff | Method | NFE | | | |
|---|---|---|---|---|---|
| | | 4 | 10 | 16 | 20 |
| CIFAR10 (discrete-time model (Ho et al., 2020), quadratic time steps) | | | | | |
| ✗ | DDIM (Song et al., 2020a) | 6.14±0.010 | 9.81±0.022 | 13.58±0.090 | 15.90±0.081 |
| ✗ | +Cache (Ma et al., 2024) | 6.31±0.150 | 13.55±0.019 | 19.42±0.091 | 24.07±0.185 |
| ✗ | +Past | 6.17±0.029 | 9.88±0.040 | 13.66±0.257 | 15.81±0.062 |
| ✗ | +Future | 6.16±0.036 | 9.77±0.153 | 13.73±0.345 | 15.67±0.096 |
| ✓ | +Past & Future | 6.10±0.006 | 9.74±0.036 | 13.48±0.220 | 15.79±0.036 |
| CelebA 64x64 (discrete-time model (Song et al., 2020a), uniform time steps) | | | | | |
| ✗ | DDIM (Song et al., 2020a) | 13.65±0.116 | 27.29±0.543 | 40.55±0.618 | 49.43±0.497 |
| ✗ | +Cache (Ma et al., 2024) | 19.82±0.130 | 45.60±0.131 | 71.70±0.266 | 89.45±0.085 |
| ✗ | +Past | 13.67±0.057 | 26.88±0.144 | 40.24±0.151 | 49.82±0.081 |
| ✗ | +Future | 13.61±0.304 | 26.38±0.067 | 39.95±0.440 | 49.05±0.543 |
| ✓ | +Past & Future | 13.21±0.060 | 26.41±0.042 | 40.26±0.186 | 49.38±0.257 |

## D.7 INCEPTION SCORE EXPERIMENTAL RESULTS

To evaluate the effectiveness of the PFDiff algorithm and the widely used Fréchet Inception Distance (FID↓) metric (Heusel et al., 2017) in the sampling process of Diffusion Probabilistic Models (DPMs), we have also incorporated the Inception Score (IS↑) metric (Salimans et al., 2016) for both unconditional and conditional pre-trained DPMs. Specifically, for the unconditional discrete-time pre-trained DPMs DDPM (Ho et al., 2020), we maintained the experimental configurations described in Table 3 of Appendix D.2, and added IS scores for the CIFAR10 dataset (Krizhevsky et al., 2009). For the unconditional continuous-time pre-trained DPMs ScoreSDE (Song et al., 2020b), the experimental configurations are consistent with Table 5 in Appendix D.3, and IS scores for the CIFAR10 dataset were also added. For the conditional classifier guidance paradigm of pre-trained DPMs ADM-G (Dhariwal & Nichol, 2021), the experimental setup aligned with Table 6 in Appendix D.4, including IS scores for the ImageNet 64x64 and ImageNet 256x256 datasets (Deng et al., 2009). Considering that the computation of IS scores relies on features extracted using `InceptionV3` pre-trained on the ImageNet dataset, calculating IS scores for non-ImageNet datasets was not feasible, hence no IS scores were provided for the classifier-free guidance paradigm of Stable-Diffusion (Rombach et al., 2022). The experimental results are presented in Table 11. A comparison between the FID↓ metrics in Tables 3, 5, and 6 and the IS↑ metrics in Table 11 shows that both IS and FID metrics exhibit similar trends under the same experimental settings, i.e., as the number of function evaluations (NFE) changes, lower FID scores correspond to higher IS scores. Further, Figs. 7a and 7b, along with the visualization experiments in Appendix D.9, demonstrate that lower FID

Table 11: Sample quality measured by IS↑ on the CIFAR10 (Krizhevsky et al., 2009), ImageNet 64x64 (Deng et al., 2009) and ImageNet 256x256 (Deng et al., 2009) using DDPM (Ho et al., 2020), ScoreSDE (Song et al., 2020b) and ADM-G (Dhariwal & Nichol, 2021) models, varying the number of function evaluations (NFE). Evaluated: ImageNet 256x256 with 10k, others with 50k samples. *We directly borrowed the results reported by AutoDiffusion (Li et al., 2023), and AutoDiffusion requires additional search costs. "\" represents missing data in the original paper and DPM-Solver-2 (Lu et al., 2022a) implementation.

| +PFDiff | Method | NFE | | | | | |
|---|---|---|---|---|---|---|---|
| | | 4 | 6 | 8 | 10 | 15 | 20 |
| CIFAR10 (discrete-time model (Ho et al., 2020), quadratic time steps) | | | | | | | |
| × | DDPM($\eta = 1.0$) (Ho et al., 2020) | 4.32 | 5.66 | 6.55 | 7.08 | 7.91 | 8.25 |
| × | Analytic-DDPM (Bao et al., 2022b) | 5.76 | 6.29 | 6.93 | 7.42 | 8.07 | 8.33 |
| × | Analytic-DDIM (Bao et al., 2022b) | 4.46 | 7.47 | 8.11 | 8.43 | 8.72 | 8.89 |
| × | DDIM (Song et al., 2020a) | 5.68 | 7.21 | 7.92 | 8.26 | 8.62 | 8.81 |
| ✓ | Analytic-DDIM | 1.62 | 8.78 | 9.43 | **9.61** | **9.35** | **9.29** |
| ✓ | DDIM | **7.79** | **9.29** | **9.62** | 9.43 | 9.29 | **9.29** |
| CIFAR10 (continuous-time model (Song et al., 2020b), quadratic time steps) | | | | | | | |
| × | DPM-Solver-1 (Lu et al., 2022a) | **7.20** | 8.30 | 8.85 | 8.98 | 9.43 | 9.51 |
| × | DPM-Solver-2 (Lu et al., 2022a) | 1.70 | 5.29 | 7.94 | 9.09 | \ | 9.74 |
| ✓ | DPM-Solver-1 | 4.29 | **9.25** | **9.76** | **9.86** | **9.85** | **9.97** |
| ✓ | DPM-Solver-2 | 6.96 | 8.58 | 8.75 | 9.26 | \ | 9.69 |
| ImageNet 64x64 (pixel DPMs model (Dhariwal & Nichol, 2021), uniform time steps, $s = 1.0$) | | | | | | | |
| × | DDIM (Song et al., 2020a) | 7.02 | 31.13 | 40.51 | 46.06 | 54.37 | 59.09 |
| × | DPM-Solver-2(Multi) (Lu et al., 2022a) | 19.03 | 33.75 | 44.65 | 51.79 | 62.18 | 67.69 |
| × | DPM-Solver-3(Multi) (Lu et al., 2022a) | 17.46 | 29.80 | 41.86 | 50.90 | 62.68 | 68.44 |
| × | DPM-Solver++(2M) (Lu et al., 2022b) | 20.72 | 34.22 | 43.62 | 50.02 | 60.00 | 65.66 |
| × | *AutoDiffusion (Li et al., 2023) | 34.88 | 43.37 | \ | 57.85 | 64.03 | 68.05 |
| ✓ | DDIM | **35.67** | **50.14** | **58.42** | **59.78** | **64.54** | **69.09** |
| ImageNet 256x256 (pixel DPMs model (Dhariwal & Nichol, 2021), uniform time steps, $s = 2.0$) | | | | | | | |
| × | DDIM (Song et al., 2020a) | 37.72 | 95.90 | 122.13 | 144.13 | 165.91 | 179.27 |
| ✓ | DDIM | **55.90** | **122.56** | **158.57** | **169.72** | **183.07** | **192.70** |

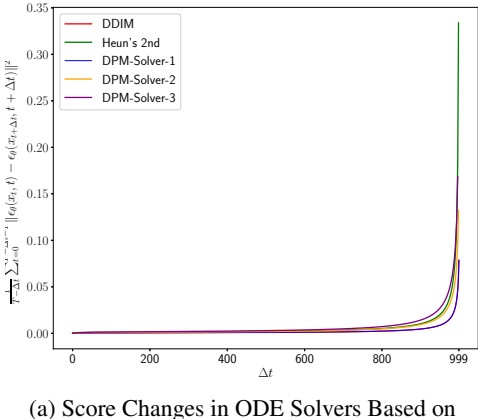 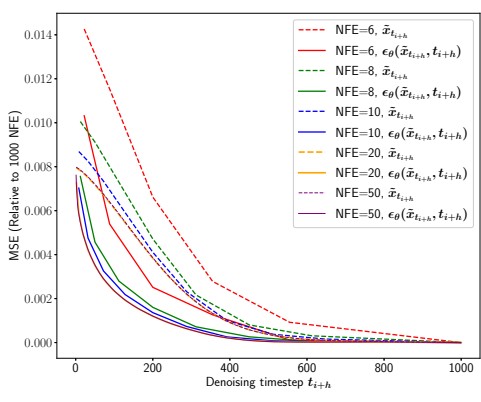

(a) Score Changes in ODE Solvers Based on EDM

(b) Future Score More Reliable than "Springboard"

Figure 6: (a) The trend of the MSE of the noise network output $\epsilon_\theta(x_t, t)$ over time step size $\Delta t$ in the EDM framework (Karras et al., 2022). (b) MSE relative to the sampling process with 1000 NFE: "Springboard", $\|\tilde{x}_{t_{i+h}} - \tilde{x}^{gt}_{t_{i+h}}\|^2$; Future score, $\|\epsilon_\theta(\tilde{x}_{t_{i+h}}, t_{i+h}) - \epsilon_\theta(\tilde{x}^{gt}_{t_{i+h}}, t_{i+h})\|^2$.

scores and higher IS scores correlate with higher image quality and richer details generated by the PFDiff sampling algorithm. These results further confirm the effectiveness of the PFDiff algorithm and the FID metric in evaluating the performance of sampling algorithms.

## D.8 EXPERIMENTAL SETUP AND ADDITIONAL RESULTS CLARIFYING THE MOTIVATION

In this section, we provide a detailed explanation of the experimental in Fig. 1a and Fig. 1b. Additionally, we further extend the experiments from Fig. 1a and Fig. 1b, as shown in Fig. 6.

For Fig. 1a, we first store the noise network output at all time steps, with 1000 NFE: $\{\epsilon_\theta(x_{t_i}, t_i)\}_{i=0}^{999}$. We then compute the mean of the MSE of the noise network output over the time interval $\Delta t$. For instance, when $\Delta t = 2$, we compute the following mean: $\|\epsilon_\theta(x_{t_0}, t_0) - \epsilon_\theta(x_{t_2}, t_2)\|^2, \cdots, \|\epsilon_\theta(x_{t_{997}}, t_{997}) - \epsilon_\theta(x_{t_{999}}, t_{999})\|^2$. By varying the value of $\Delta t$ from 0 to 999, we are able to compute the mean of the MSE of the noise network output, respectively, and ultimately obtain Fig. 1a. Notably, in Fig. 1a, the curves for DDPM (Ho et al., 2020) and DDIM (Song et al., 2020a) are derived from the pre-trained model of DDPM (Ho et al., 2020); the curves for DPM-Solver (Lu et al., 2022a) and DPM-Solver++ (Lu et al., 2022b) are obtained from the pre-trained model of ScoreSDE (Song et al., 2020b). Additionally, in Fig. 6a, we present experimental results from a more advanced diffusion model framework — the EDM (Karras et al., 2022) pre-trained model, including DDIM, DPM-Solver, and Heun's 2nd (Karras et al., 2022) solvers. For the three different architecture pre-trained models, when the time step size $\Delta t$ is not excessively large, the noise network outputs exhibit remarkable similarity. This validates that the method we propose in Sec. 3.3, using past scores to guide sampling, is reliable.

For Fig. 1b, during the sampling process from $\tilde{x}_{t_i}$ to $\tilde{x}_{t_{i+(k+1)}}$, based on the PFDiff, we first compute the "springboard" $\tilde{x}_{t_{i+h}}$, and then further obtain the future score $\epsilon_\theta(\tilde{x}_{t_{i+h}}, t_{i+h})$. Next, we compute the MSE of the status $\tilde{x}_{t_{i+(k+1)}}$, which is updated using the "springboard" and future score, respectively. The MSE is calculated relative to the sampling process with 1000 NFE, resulting in Fig. 1b. Moreover, We also directly evaluate the MSE of the "springboard" and the future score at different $t_{i+h}$ moments, relative to the sampling with 1000 NFE, as shown in Fig. 6b. Notably, both Fig. 1b and Fig. 6b exhibit the same trend, demonstrating that the future gradient is more reliable than the "Springboard".

## D.9 VISUALIZE STUDY RESULTS

To demonstrate the effectiveness of PFDiff, we present the visual sampling results on the CI-FAR10 (Krizhevsky et al., 2009), CelebA 64x64 (Liu et al., 2015), LSUN-bedroom 256x256 (Yu et al., 2015), LSUN-church 256x256 (Yu et al., 2015), ImageNet 64x64 (Deng et al., 2009), Ima-

geNet 256x256 (Deng et al., 2009), and MS-COCO2014 (Lin et al., 2014) datasets in Figs. 7-13. These results illustrate that PFDiff, using different orders of ODE solvers as a baseline, is capable of generating samples of higher quality and richer detail on both unconditional and conditional pre-trained Diffusion Probabilistic Models (DPMs).

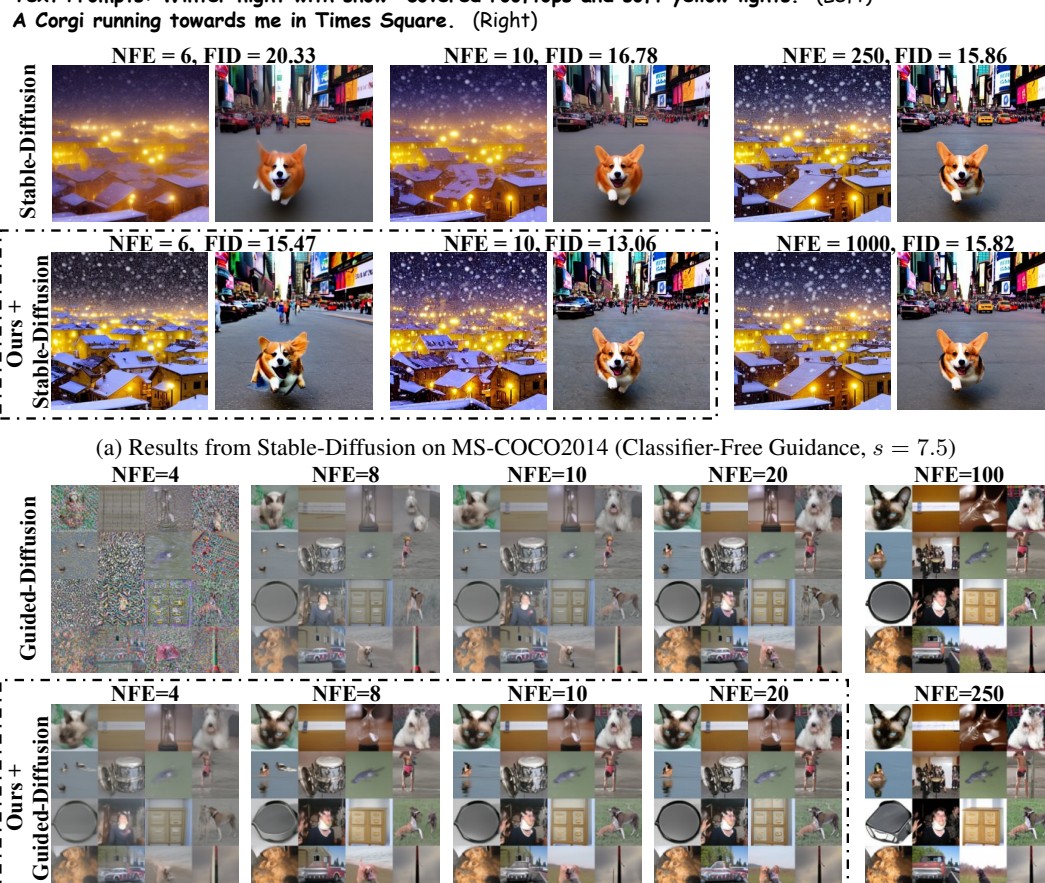

(a) Results from Stable-Diffusion on MS-COCO2014 (Classifier-Free Guidance, $s = 7.5$)

(b) Results from Guided-Diffusion on ImageNet 64x64 (Classifier Guidance, $s = 1.0$)

Figure 7: Sampling by conditional pre-trained DPMs (Rombach et al., 2022; Dhariwal & Nichol, 2021) using DDIM (Song et al., 2020a) and our method PFDiff (dashed box) with DDIM as a baseline, varying the number of function evaluations (NFE).

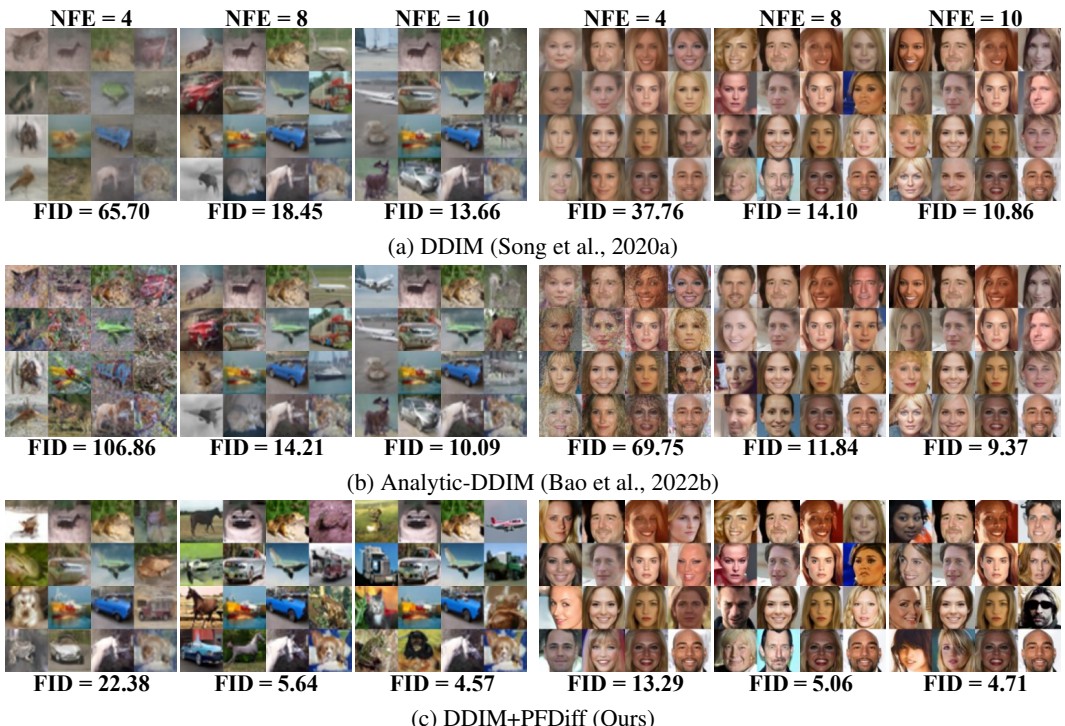

Figure 8: Random samples by DDIM (Song et al., 2020a), Analytic-DDIM (Bao et al., 2022b), and PFDiff (baseline: DDIM) with 4, 8, and 10 number of function evaluations (NFE), using the same random seed, quadratic time steps, and pre-trained discrete-time DPMs (Ho et al., 2020; Song et al., 2020a) on CIFAR10 (Krizhevsky et al., 2009) (left) and CelebA 64x64 (Liu et al., 2015) (right).

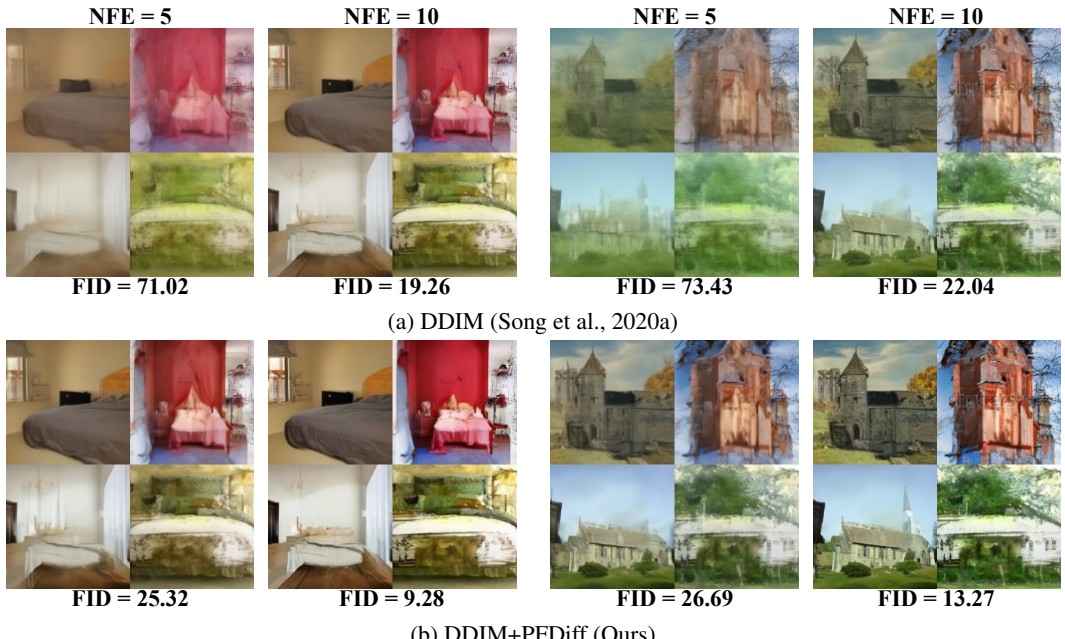

Figure 9: Random samples by DDIM (Song et al., 2020a) and PFDiff (baseline: DDIM) with 5 and 10 number of function evaluations (NFE), using the same random seed, uniform time steps, and pre-trained discrete-time DPMs (Ho et al., 2020) on LSUN-bedroom 256x256 (Yu et al., 2015) (left) and LSUN-church 256x256 (Yu et al., 2015) (right).

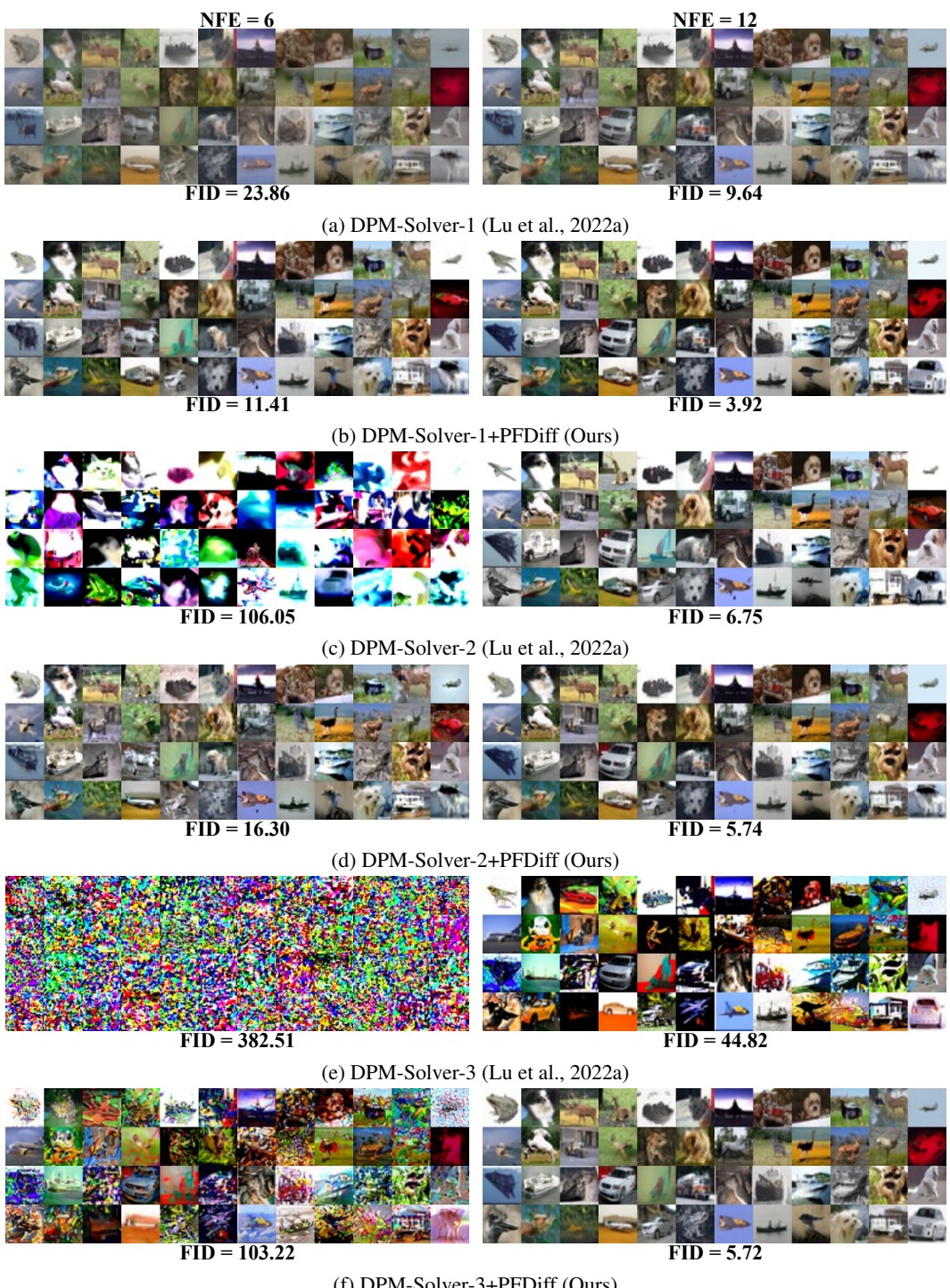

Figure 10: Random samples by DPM-Solver-1, -2, and -3 (Lu et al., 2022a) with and without our method (PFDiff) with 6 and 12 number of function evaluations (NFE), using the same random seed, quadratic time steps, and pre-trained continuous-time DPMs (Song et al., 2020b) on CIFAR10 (Krizhevsky et al., 2009).

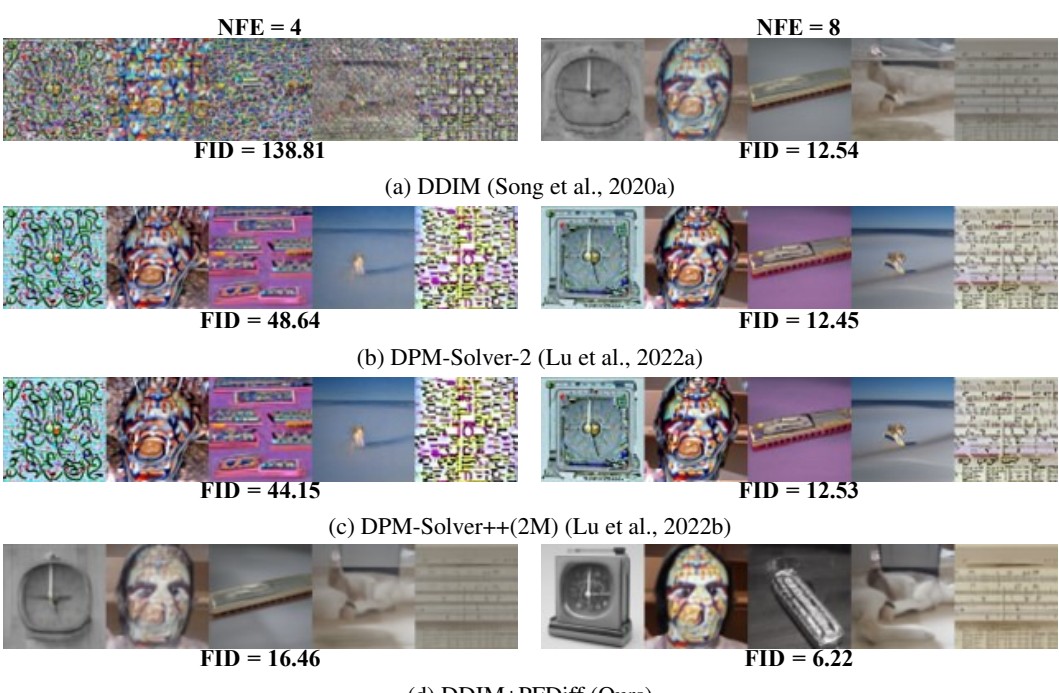

Figure 11: Random samples by DDIM (Song et al., 2020a), DPM-Solver-2 (Lu et al., 2022a), DPM-Solver++(2M) (Lu et al., 2022b), and PFDiff (baseline: DDIM) with 4 and 8 number of function evaluations (NFE), using the same random seed, uniform time steps, and pre-trained Guided-Diffusion (Dhariwal & Nichol, 2021) on ImageNet 64x64 (Deng et al., 2009) with a guidance scale of 1.0.

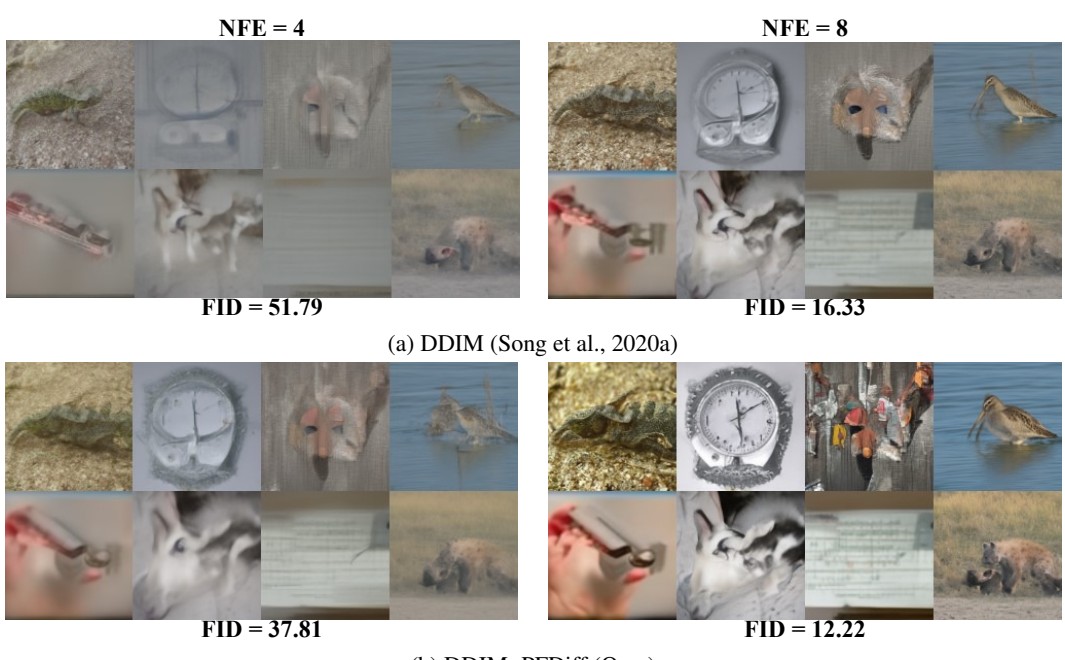

Figure 12: Random samples by DDIM (Song et al., 2020a) and PFDiff (baseline: DDIM) with 4 and 8 number of function evaluations (NFE), using the same random seed, uniform time steps, and pre-trained Guided-Diffusion (Dhariwal & Nichol, 2021) on ImageNet 256x256 (Deng et al., 2009) with a guidance scale of 2.0.

**Text Prompts** (listed from left to right):
A large bird is standing in the water by some rocks.
A candy covered cup cake sitting on top of a white plate.
People at a wine tasting with a table of wine bottles and glasses of red wine.
A bathtub sits on a tiled floor near a sink that has ornate mirrors over it while greenery grows on the other side of the tub.
A kitchen and dining area in a house with an open floor plan that looks out over the landscape from a large set of windows.

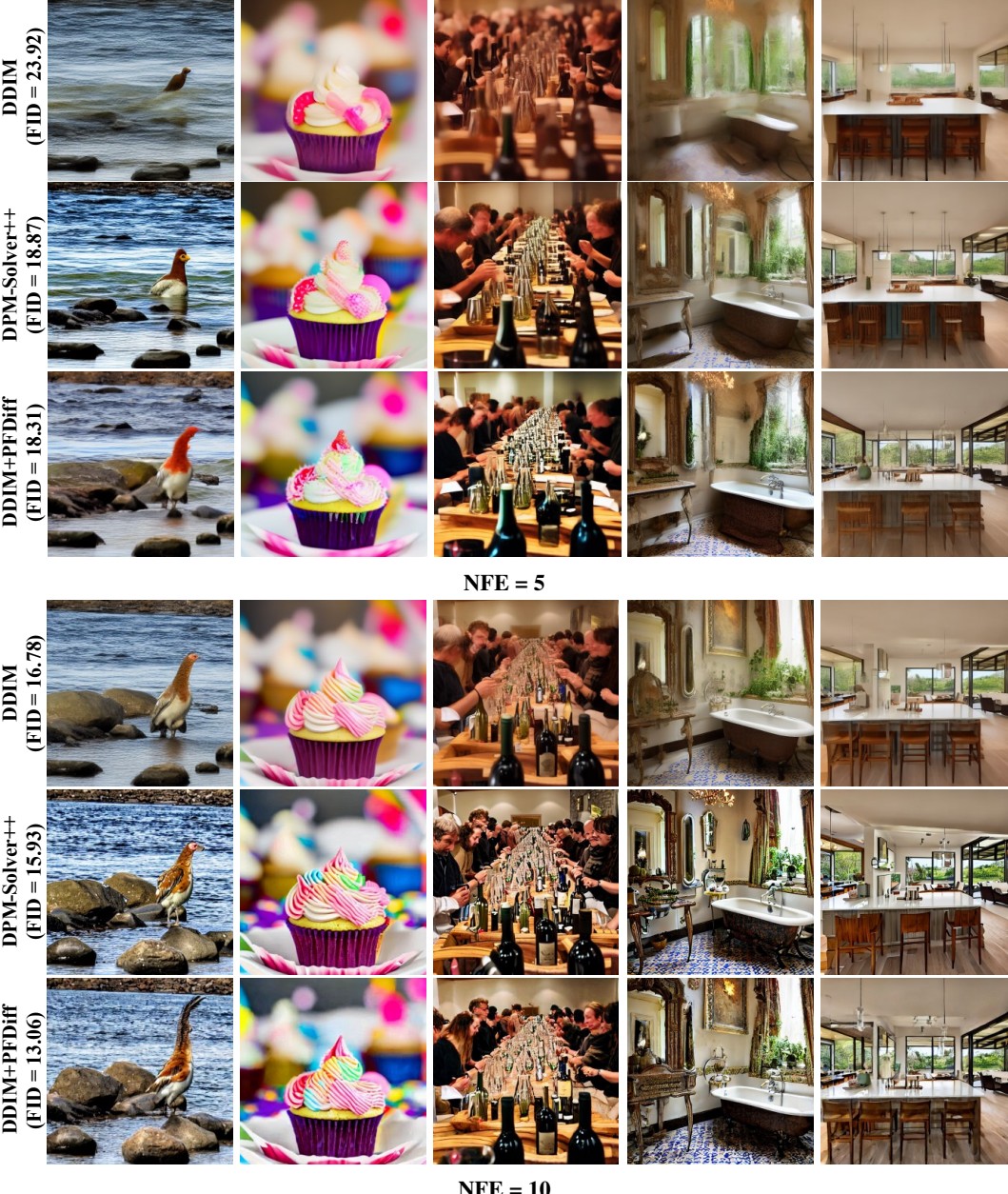

Figure 13: Random samples by DDIM (Song et al., 2020a), DPM-Solver++(2M) (Lu et al., 2022b), and PFDiff (baseline: DDIM) with 5 and 10 number of function evaluations (NFE), using the same random seed, uniform time steps, and pre-trained Stable-Diffusion (Rombach et al., 2022) with a guidance scale of 7.5. Text prompts are a random sample from the MS-COCO2014 (Lin et al., 2014) validation set.

