# OpenReview forum: "PFDiff: Training-Free Acceleration of Diffusion Models Combining Past and Future Scores"
_ICLR.cc/2025/Conference — ICLR 2025 Poster_

### Official Review · Reviewer_7KLd · 2024-10-25

**Soundness:** 3
**Presentation:** 2
**Contribution:** 3
**Rating:** 6
**Confidence:** 4

**Summary:**

This paper introduces a sampling method to accelerate the first-order ODE solvers by utilizing the past gradient ($\epsilon$) and future gradient, leading to FID improvements in the NFE regime between 4 and 20. Also, the authors accelerate the higher-order ODE solver by using the past gradients. Empirical results show that the sampling method works well for continuous and discrete diffusion models under unconditional and conditional settings.

**Strengths:**

1. Unlike other generic ODE solvers (e.g., Heun 2nd-order solver, DPM-solver), this paper proposes the reuse of the network output from previous time steps to accelerate the sampling based on the observation of the output similarity between two consecutive time steps.
2. Experiments on various diffusion models (continuous, discrete, conditional, unconditional) verify that PFdiff-1 outperforms other ODE solvers in the low NFE regime (4-20)

**Weaknesses:**

1. The proposed method lacks theoretical support and is mainly motivated by the output similarity observation shown in Figure 2a. How reliable is this similarity? How much does this observation vary in different diffusion frameworks? Please provide more details of experiments in Figures 2a and 2b, you can put the details in into appendix. I suggest the authors explore the connection between the sampling trajectory and the proposed sampling method. I think the curvature of the trajectory can explain the reuse of the gradient and your methods. Refer to [1] and [2] for details of trajectory shape.

2. the overall writing is problematic and significantly affects the readability of this paper. I list some below:
- the definition of Q is not clear, in line 222, plug in n=0 does not give $x_{t_{i-1}}$. Please rethink the expression of Q since it is used throughout the paper.
- if the proposal of using future gradients is based on Proposition 3.1, why not put the proposition at the beginning of section 3.3?
- function s() is not defined in eq 7 and eq 8
- function h() is not defined in eq 9
- line 348, notations l and h are undefined
- In Figure 2b, treating the samples derived from 1000NFE as the ground truth is not rigorous.

3. The authors claim that PFDiff is effective and orthogonal to existing ODE solvers, please provide the FID results of PFDiff in the regime NFE>20 to support the claim.

4. in Figure 4, some FID results of PFDiff are missing (NFE=4 and NFE>12). In Figure 5, some FID results of PFDiff are missing (NFE>10)

5. in Figure 4b, why PFDiff is worse than the baseline Analytic-DDIM when NFE=6? A similar outlier in Figure 4a

6. in Figure 5, the results of DPM-solver+PFDiff are missing.

7. I encourage the authors to also compare the FID of PFDiff with [2]


others:
1. line 52, the last two papers are published in 2023, not 2024, please cite papers correctly
2. I suggest the authors move Figure 1 to the appendix to leave space for the main content.


[1] Sabour, Amirmojtaba, Sanja Fidler, and Karsten Kreis. "Align your steps: Optimizing sampling schedules in diffusion models." ICML. 2024

[2] Zhou, Zhenyu, et al. "Fast ode-based sampling for diffusion models in around 5 steps." CVPR. 2024

**Questions:**

have the authors also tried adding the future gradient step to higher-order ODE solvers?

---

> ### Author Response · Authors · 2024-11-17
>
> We sincerely appreciate the reviewer for carefully reviewing our work and providing valuable comments. Below are our responses to all questions. We would greatly appreciate it if you could consider increasing the score if you are satisfied with our response.
>
> ***W1.1: The proposed method lacks theoretical support and is mainly motivated by the output similarity observation shown in Figure 2a. How reliable is the similarity observed in Figure 2a, and how much does it vary across different diffusion frameworks?***
>
> **A:** It is important to emphasize that our method consists of two core components: the use of future gradients and the use of past gradients. In Equation (15) and Proposition 3.1, we employ a Taylor expansion and further derivations to theoretically demonstrate that the use of future gradients can reduce discretization errors. Furthermore, the use of past gradients, based on the similarity observation in Figure 1a (original Figure 2a), is highly reliable. We have supplemented additional experimental results in Figure 5a and further analysis in Appendix D.8. The results demonstrate that this similarity exhibits strong consistency across three different diffusion frameworks: DDPM, ScoreSDE [R1], and EDM [R2], which fully validates the reliability of the similarity observation.
>
> ***W1.2: Please provide more details of experiments in Figures 2a and 2b.***
>
> **A:** We have provided more details of the experiments regarding Figures 1a (original Figure 2a) and 1b (original Figure 2b) in Appendix D.8. Additionally, we added Figure 5 in Appendix D.8 to further expand the experimental content in this part. Figure 5a demonstrates that the similarity observation is consistent across different diffusion frameworks, while Figure 5b further validates that future gradient is more reliable than "springboard".
>
> **W1.3: Please explore the connection between the proposed method and the curvature of the sampling trajectory.**
>
> **A1:** Thank you for your valuable suggestions. We have carefully reviewed the literature on sampling trajectories [R3], [R4], [R5] and have added relevant analysis to Section 3.5. In summary, [R4] indicates that diffusion model sampling trajectories lie in a low-dimensional subspace and are close to a straight line. This supports the rationale behind the use of past gradients. [R5] notes that the trajectories exhibit a "boomerang" shape, meaning the curvature of the sampling trajectory starts small, then increases, and finally decreases. Based on this observation, we further analyze that PFDiff utilizes future gradients to predict the future update direction in the large curvature regions of the trajectory, thereby correcting discretization errors introduced by sampling along the tangent direction. Additionally, the PFDiff correction process shown in Figure 1c (original Figure 2c) is consistent with this analysis.
>
> ***W2: Regarding writing issues.***
>
> **A:** We sincerely thank the reviewer for highlighting the writing issues. While we respectfully acknowledge the concerns raised, we have a differing perspective on some points and will address each one individually.
>
> - **The definition of $Q$ is not clear.** We corrected the definition of $Q$ on lines 183 to 184. Specifically, when $n=0$, $\hat{t}\_{0} = t\_{i-1}$.
> - **Why not put Proposition 3.1 at the beginning of section 3.3?** We made this arrangement to ensure that the logical description of the PFDiff update process in Sections 3.2 and 3.3 is more coherent.
> - **Function $s()$ is not defined in eq 7 and eq 8.** In fact, the function $s()$ is already defined in eq (7). Specifically, $s(\epsilon_\theta(x_t,t),x_t,t) :=f(t)x_t+\frac{g^2(t)}{2\sigma_t}\epsilon_\theta(x_t,t)$.
> - **Function $h()$ is not defined in eq 9.** In fact, the function $h()$ is already defined in lines 185 to 189.
> - **Line 348, notations $l$ and $h$ are undefined.** We added the definition of $h$ in lines 327 to 328. Notably, the "1" is the digit "1" (not the letter $l$).
> - **Using 1000 NFE as the ground truth is not rigorous.** We removed the expression of "ground truth" in Figure 1b (original Figure 2b).
>
> [R1] Yang Song et al., "Score-based generative modeling through stochastic differential equations." ICLR 2021.
>
> [R2] Tero Karras et al., "Elucidating the design space of diffusion-based generative models." NeurIPS 2022.
>
> [R3] Sabour, Amirmojtaba, Sanja Fidler, and Karsten Kreis. "Align your steps: Optimizing sampling schedules in diffusion models." ICML 2024.
>
> [R4] Zhou, Zhenyu, et al. "Fast ode-based sampling for diffusion models in around 5 steps." CVPR 2024.
>
> [R5] Defang Chen, et al. "On the Trajectory Regularity of ODE-based Diffusion Sampling." ICML 2024.

---

> ### Author Response · Authors · 2024-11-17
>
> ***W3: Additional FID results of PFDiff for NFE > 20.***
>
> We supplemented the FID experimental results for NFE > 20 on the CIFAR10 and CelebA 64$\times$64 datasets, as shown in the table below:
>
> CIFAR10, FID$\downarrow$
>
> | Method\NFE  | 20       | 50       | 100      | 200      | 1000 |
> | ----------- | -------- | -------- | -------- | -------- | ---- |
> | DDIM        | 7.04     | 4.81     | 4.30     | 4.00     | 4.04 |
> | DDIM+PFDiff | **3.68** | **3.48** | **3.56** | **3.57** | \    |
>
> CelebA 64$\times$64, FID$\downarrow$
>
> | Method\NFE  | 20       | 50       | 100      | 200      | 1000 |
> | ----------- | -------- | -------- | -------- | -------- | ---- |
> | DDIM        | 14.76    | 9.22     | 6.57     | 4.77     | 3.51 |
> | DDIM+PFDiff | **5.66** | **3.99** | **2.45** | **2.58** | \    |
>
> PFDiff is designed for diffusion model acceleration, which is the reason we focus on less NFE. Still, as can be seen from the table, under the condition of NFE > 20, PFDiff consistently improves the sampling quality of DDIM and achieves faster convergence.
>
> ***W4: Some FID results of PFDiff are missing.***
>
> **A:** In fact, the missing FID results for PFDiff were originally provided in Tables 2-6 of Appendix D. These results were deliberately omitted to make the comparisons in the figures clearer. For example, when NFE=12 in Figure 3 (original Figure 4) and NFE=10 in Figure 4 (original Figure 5), PFDiff already outperforms the comparison methods with 20 NFE. Therefore, omitting the subsequent PFDiff results helps to highlight the effectiveness of our method.
>
> **W5: Why PFDiff is worse than the baseline Analytic-DDIM when NFE=6?**
>
> **A:** This phenomenon may be due to the introduction of new errors in the estimation of the optimal variance by Analytic-DDIM [R1], which are amplified by PFDiff when NFE is small, leading to some unpredictable performance degradation. Specifically, although Analytic-DDIM derives the optimal variance for DDIM sampling from a theoretical perspective, the optimal variance contains an unknown term: $\mathbb{E}\_{q\_{n}\left(\boldsymbol{x}\_{n}\right)} \frac{\| \| \nabla\_{\boldsymbol{x}\_{n}} \log q\_{n}\left(\boldsymbol{x}\_{n}\right)\| \|^{2}}{d}$, which is approximated on different datasets [R1]. This introduces new errors.
>
> ***W6: In Figure 5, the results of DPM-solver+PFDiff are missing.***
>
> **A:** We did not include the experimental results for the following reasons: First, DDIM and DPM-Solver-1 are equivalent [R2]. Second, with fewer NFE, higher-order DPM-Solvers perform worse than DDIM on Stable-Diffusion [R3]. Finally, DDIM+PFDiff has outperformed other previously best-performing solvers. Additionally, we also provide experimental results applying PFDiff to DPM-Solver in Stable Diffusion with a guidance scale of 7.5, as shown in the table below:
>
> Stable Diffusion, FID$\downarrow$
>
> | Method\NFE          | 4         | 6         | 8         | 10        | 15        | 20        |
> | ------------------- | --------- | --------- | --------- | --------- | --------- | --------- |
> | DPM-Solver-1 [R2]   | 35.48     | 20.33     | 17.46     | 16.78     | 16.08     | 15.95     |
> | DPM-Solver-1+PFDiff | **29.02** | **15.47** | **13.26** | **13.06** | **13.57** | **13.97** |
> | DPM-Solver-2 [R2]   | 184.21    | 157.95    | 148.67    | 135.81    | 92.62     | 40.47     |
> | DPM-Solver-2+PFDiff | 147.20    | 106.24    | 57.07     | 31.66     | 17.87     | 14.13     |
>
> As shown in the table above, PFDiff further improves the sampling quality of DPM-Solver-1 and DPM-Solver-2. Notably, the results in the table come from the original DPM-Solver implementation, while the DPM-Solver results in the paper utilized additional tricks from [R3] to achieve better performance.
>
> [R1] Fan Bao, et al. "Analytic-dpm: an analytic estimate of the optimal reverse variance in diffusion probabilistic models.", ICLR 2022.
>
> [R2] Cheng Lu et al., “DPM-solver: a fast ODE solver for diffusion probabilistic model sampling in around 10 steps.” NeurIPS 2022.
>
> [R3] Cheng Lu et al., "Dpm-solver++: Fast solver for guided sampling of diffusion probabilistic models." arXiv:2211.01095.

---

> ### Author Response · Authors · 2024-11-17
>
> ***W7: The comparison of FID between PFDiff and AMED-Solver [R1].***
>
> **A:** In response to the reviewer's suggestion, we have included additional FID comparison results between PFDiff and AMED-Solver [R1]. Notably, [R1] is a TRAINING-BASED method, while PFDiff is a TRAINING-FREE method. Using the same pre-trained EDM model [R2], we calculated the FID on the ImageNet 64$\times$64 dataset, as shown in the following table:
>
> ImageNet 64$\times$64, FID$\downarrow$
>
> | Method\NFE            | 4         | 6         | 8        | 10       |
> | --------------------- | --------- | --------- | -------- | -------- |
> | DDIM                  | 58.43     | 34.03     | 22.59    | 16.72    |
> | DDIM+AMED-Plugin [R1] | 41.80     | 32.49     | 22.04    | 16.60    |
> | AMED-Solver [R1]      | **30.03** | 19.24     | 9.58     | 6.60     |
> | DDIM+PFDiff           | 55.63     | **17.82** | **8.00** | **5.38** |
>
> The results show that the proposed PFDiff, which incurs no training cost, can even outperform AMED-Solver [R1], one of the current strongly competitive training-based methods, under most NFE. This further demonstrates the effectiveness of the proposed PFDiff.
>
> ***Others: Correct the reference years and move Figure 1 to the appendix.***
>
> **A:** Thank you for the reviewer's comment. We have moved Figure 1 to Appendix D.9 as suggested and have corrected the reference year in line 52.
>
> ***Q1: Have the authors also tried adding the future gradient step to higher-order ODE solvers?***
>
> **A:** Yes, we have. However, adding the future gradient into higher-order ODE solvers is highly sensitive to the choices of hyperparameters $k$ and $h$, leading to unstable results. We believe this instability may arise from the fact that higher-order solvers do not converge with a few NFE (<10) [R3].
>
> [R1] Zhou, Zhenyu, et al. "Fast ode-based sampling for diffusion models in around 5 steps." CVPR 2024.
>
> [R2] Tero Karras et al., "Elucidating the design space of diffusion-based generative models." NeurIPS 2022.
>
> [R3] Cheng Lu et al., "DPM-solver: a fast ODE solver for diffusion probabilistic model sampling in around 10 steps." NeurIPS 2022.

---

> > ### Comment · Reviewer_7KLd · 2024-11-19
> >
> > I appreciate the authors' effort in addressing my concerns, I have increased my rating since some of my doubts are resolved.
> >
> > I like the idea of using past and future scores. However, the proposed ODE solver PFDiff-k_h still has the issue of empirically searching for the optimal k and h in practice. Table 7 in the appendix shows that the sample quality (FID) is sensitive to the choice of k,h in different datasets and NFE.
> >
> > While the authors acknowledge the correlation between the trajectory shape of the diffusion process and the performance of PFDiff-k_h, I suggest that further investigation into a systematic method for determining k and h is very beneficial, elucidating the underlying mechanics of the proposed method more clearly. It could also enhance the robustness and efficiency of the solver.

---

> ### Author Response · Authors · 2024-11-21
>
> Thank you for your response and for raising the rating! We are so glad to hear that some of your doubts have been resolved!
>
> Regarding the new issue about the search for parameters $k$ and $h$, we emphasize that the sample quality is **only slightly affected** by the choice of $k$ and $h$. By fixing $k=2$ and $h=1$, PFDiff achieves fairly satisfactory results across different NFE and datasets, as shown in Table 7. Furthermore, in Appendix D.6.1, we have proposed that by sampling **only 5k** samples to compute the FID, the optimal values of $k$ and $h$ can be easily determined. Notably, the search can also be easily generalized to other evaluation metrics. As suggested by the reviewer, we further analyzed the trajectory shape of the diffusion process and proposed **a new Automatic Search strategy** with almost no additional cost based on truncation errors for selecting $k$ and $h$. The details are as follows:
>
> For the trajectory shape analysis, as discussed in Section 3.5, PFDiff uses future scores to correct the discretization errors of the baseline solver in regions of large curvature along the trajectory. This indicates that PFDiff accumulates less truncation error compared to the baseline solver. To validate this conclusion, we **added Figure 5 in Appendix D.6.1**, which shows a comparison of the truncation errors of PFDiff with and without DDIM. Figure 5 clearly shows that PFDiff significantly reduces truncation errors.
>
> Based on the correlation between PFDiff's improvement in sample quality and its reduction in truncation errors, we propose the automatic search strategy: we "warm up" **only 256 samples** for different $k$ and $h$ respectively, compute the average truncation error using MSE for each, and use this to further determine the values of $k$ and $h$. To validate the effectiveness of this strategy, we conducted experiments on CIFAR10, varying the values of NFE, $k$, and $h$. For each combination, we sampled 256 samples and computed the average truncation error relative to DDIM with 1000 NFE. The results are shown in Table A1 below (for clearer comparison, we also provide the FID results with 50k samples in Table A2 (Table 7 in the paper)):
>
> Table A1: Truncation error (MSE$\downarrow$), only 256 samples
>
> | Method\NFE  | 4          | 6          | 8          | 10         | 12         | 15         | 20         |
> | ----------- | ---------- | ---------- | ---------- | ---------- | ---------- | ---------- | ---------- |
> | DDIM        | 0.1009     | 0.0608     | 0.0414     | 0.0314     | 0.0255     | 0.0199     | 0.0152     |
> | +PFDiff-1   | 0.0542     | 0.0217     | 0.0131     | **0.0100** | **0.0089** | **0.0082** | **0.0081** |
> | +PFDiff-2_1 | 0.0277     | **0.0137** | **0.0110** | 0.0104     | 0.0098     | 0.0093     | 0.0088     |
> | +PFDiff-2_2 | 0.1001     | 0.0468     | 0.0277     | 0.0184     | 0.0145     | 0.0122     | 0.0105     |
> | +PFDiff-3_1 | **0.0218** | 0.0167     | 0.0146     | 0.0130     | 0.0120     | 0.0107     | 0.0098     |
> | +PFDiff-3_2 | 0.0614     | 0.0228     | 0.0133     | 0.0101     | **0.0089** | 0.0083     | 0.0082     |
> | +PFDiff-3_3 | 0.1790     | 0.0820     | 0.0444     | 0.0299     | 0.0224     | 0.0165     | 0.0126     |
>
> Table A2: FID$\downarrow$, 50k samples
>
> | Method\NFE  | 4         | 6        | 8        | 10       | 12       | 15       | 20       |
> | ----------- | --------- | -------- | -------- | -------- | -------- | -------- | -------- |
> | DDIM        | 65.70     | 29.68    | 18.45    | 13.66    | 11.01    | 8.80     | 7.04     |
> | +PFDiff-1   | 124.73    | 19.45    | 5.78     | 4.95     | 4.63     | 4.25     | 4.14     |
> | +PFDiff-2_1 | 59.61     | **9.84** | 7.01     | 6.31     | 5.58     | 5.18     | 4.78     |
> | +PFDiff-2_2 | 167.12    | 53.22    | 8.43     | 4.95     | 4.41     | **4.10** | 3.78     |
> | +PFDiff-3_1 | **22.38** | 13.40    | 9.40     | 7.70     | 6.73     | 6.03     | 5.05     |
> | +PFDiff-3_2 | 129.18    | 19.35    | **5.64** | **4.57** | **4.39** | 4.19     | 4.08     |
> | +PFDiff-3_3 | 205.87    | 76.62    | 20.84    | 5.71     | 4.73     | 4.41     | **3.68** |
>
> Based on Table A1, we selected the $k$ and $h$ corresponding to the minimal truncation error, and further obtained the corresponding FID values {22.38, 9.84, 7.01, 4.95, 4.63/4.39, 4.25, 4.14} with NFE $\in${4, 6, 8, 10, 12, 15, 20} in Table A2. This set of FID values is comparable to the optimal FID, indicating that determining $k$ and $h$ based on truncation error is reasonable. Notably, in Table A1, we "warmed up" only 256 samples, so the additional cost introduced by the automatic search strategy is negligible. The 256 samples are sufficient to capture the dataset's statistical properties because the shapes of sampling trajectories are highly similar [R1]. The above-mentioned relevant results and analyses have been added to lines 477-479 of Section 4.3 and Appendix D.6.1.
>
> [R1] Defang Chen, et al. "On the Trajectory Regularity of ODE-based Diffusion Sampling." ICML 2024.

---

> > ### Comment · Reviewer_7KLd · 2024-11-23
> >
> > Thank you again for the detailed response and great effort.
> >
> > Could you extend the automatic strategy experiments to additional datasets beyond CIFAR-10 to demonstrate its robustness?
> >
> > I will undoubtedly raise my rating if the automatic search shows good generalization.

---

> ### Author Response · Authors · 2024-11-24
>
> Thank you again for your response and insightful suggestions!
>
> Per the reviewer’s suggestion, we have extended the automatic search strategy to additional datasets, including CelebA 64$\times$64, LSUN Church 256$\times$256, and LSUN Bedroom 256$\times$256. Specifically, we "warm up" **only 256 samples** for different values of $k$, $h$, and NFE, respectively, and computed the average truncation error using MSE relative to DDIM with 1000 NFE. The results are presented in Tables A1, A3, and A5 below. For clearer comparison, we also provide the FID results in Tables A2, A4, and A6. Based on Tables A1, A3, and A5, we selected the values of $k$ and $h$ corresponding to the minimal truncation error, and further obtained the corresponding FID values {13.29, 8.38, 5.88, 5.41, 5.24, 5.18, 5.19}, {37.90, 18.30, 14.34, 13.41, 12.90, 12.61, 11.76}, and {78.57, 18.72, 12.35, 9.55, 8.82, 8.34, 7.70} from Tables A2, A4, and A6, with NFE $\in$ {4/5, 6, 8, 10, 12, 15, 20}. Consistent with CIFAR-10, this set of FID values is comparable to the optimal FID, indicating that the automatic search strategy is robust and generalizable.
>
> Table A1: CelebA 64$\times$64, Truncation error (MSE$\downarrow$), only 256 samples
>
> | Method\NFE  | 4          | 6          | 8          | 10         | 12         | 15         | 20         |
> | ----------- | ---------- | ---------- | ---------- | ---------- | ---------- | ---------- | ---------- |
> | DDIM        | 0.0814     | 0.0614     | 0.0494     | 0.0426     | 0.0388     | 0.0357     | 0.0336     |
> | +PFDiff-1   | 0.0689     | 0.0427     | 0.0354     | 0.0328     | 0.0322     | 0.0323     | 0.0329     |
> | +PFDiff-2_1 | 0.0470     | 0.0339     | **0.0312** | **0.0310** | **0.0311** | **0.0316** | 0.0321     |
> | +PFDiff-2_2 | 0.1184     | 0.0578     | 0.0424     | 0.0378     | 0.0362     | 0.0357     | 0.0354     |
> | +PFDiff-3_1 | **0.0389** | **0.0321** | 0.0315     | 0.0315     | 0.0316     | 0.0318     | **0.0320** |
> | +PFDiff-3_2 | 0.0903     | 0.0402     | 0.0338     | 0.0327     | 0.0324     | 0.0329     | 0.0331     |
> | +PFDiff-3_3 | 0.1859     | 0.0808     | 0.0521     | 0.0440     | 0.0406     | 0.0389     | 0.0375     |
>
> Table A2: CelebA 64$\times$64, FID$\downarrow$, 50k samples
>
> | Method\NFE  | 4         | 6        | 8        | 10       | 12       | 15       | 20       |
> | ----------- | --------- | -------- | -------- | -------- | -------- | -------- | -------- |
> | DDIM        | 37.76     | 20.99    | 14.10    | 10.86    | 9.01     | 7.67     | 6.50     |
> | +PFDiff-1   | 90.60     | 24.94    | 5.49     | 4.88     | 4.71     | **4.70** | 4.72     |
> | +PFDiff-2_1 | 48.66     | **7.53** | 5.88     | 5.41     | 5.24     | 5.18     | 4.97     |
> | +PFDiff-2_2 | 221.52    | 53.55    | 10.71    | 5.41     | 4.90     | 4.79     | 4.79     |
> | +PFDiff-3_1 | **13.29** | 8.38     | 7.03     | 6.33     | 5.92     | 5.73     | 5.19     |
> | +PFDiff-3_2 | 131.02    | 22.63    | **5.06** | **4.71** | **4.60** | **4.70** | **4.68** |
> | +PFDiff-3_3 | 286.12    | 100.7    | 22.95    | 6.55     | 5.61     | 5.21     | 5.08     |

---

> > ### Author Response · Authors · 2024-11-24
> >
> > Table A3: LSUN Church 256$\times$256, Truncation error (MSE$\downarrow$), only 256 samples
> >
> > | Method\NFE  | 5          | 6          | 8          | 10         | 12         | 15         | 20         |
> > | ----------- | ---------- | ---------- | ---------- | ---------- | ---------- | ---------- | ---------- |
> > | DDIM        | 0.0772     | 0.0621     | 0.0444     | 0.0337     | 0.0269     | 0.0206     | 0.0146     |
> > | +PFDiff-1   | 0.5523     | 0.3391     | 0.0598     | 0.0345     | 0.0232     | 0.0136     | 0.0068     |
> > | +PFDiff-2_1 | **0.0256** | **0.0195** | 0.0141     | 0.0111     | 0.0091     | 0.0071     | 0.0050     |
> > | +PFDiff-2_2 | 0.8345     | 0.6097     | 0.2709     | 0.0662     | 0.0424     | 0.0251     | 0.0131     |
> > | +PFDiff-3_1 | 0.0323     | 0.0268     | 0.0192     | 0.0151     | 0.0122     | 0.0095     | 0.0067     |
> > | +PFDiff-3_2 | 0.2108     | 0.0456     | **0.0140** | **0.0103** | **0.0078** | **0.0055** | **0.0033** |
> > | +PFDiff-3_3 | 0.9681     | 0.7627     | 0.4034     | 0.1033     | 0.0584     | 0.0350     | 0.0183     |
> >
> > Table A4: LSUN Church 256$\times$256, FID$\downarrow$, 50k samples
> >
> > | Method\NFE  | 5         | 6         | 8         | 10        | 12        | 15        | 20        |
> > | ----------- | --------- | --------- | --------- | --------- | --------- | --------- | --------- |
> > | DDIM        | 73.43     | 50.02     | 30.04     | 22.04     | 17.66     | 14.58     | 12.49     |
> > | +PFDiff-1   | 214.89    | 143.60    | 55.00     | 38.01     | 28.95     | 21.80     | 16.48     |
> > | +PFDiff-2_1 | 37.90     | **18.30** | 14.35     | **13.27** | 12.47     | 12.25     | 11.64     |
> > | +PFDiff-2_2 | 277.36    | 231.01    | 113.18    | 47.71     | 35.47     | 26.00     | 18.64     |
> > | +PFDiff-3_1 | **26.69** | 21.19     | 15.48     | 13.36     | **12.05** | **11.77** | **11.12** |
> > | +PFDiff-3_2 | 169.02    | 54.35     | **14.34** | 13.41     | 12.90     | 12.61     | 11.76     |
> > | +PFDiff-3_3 | 313.39    | 274.65    | 116.95    | 56.95     | 39.80     | 28.90     | 19.82     |
> >
> >
> > Table A5: LSUN Bedroom 256$\times$256, Truncation error (MSE$\downarrow$), only 256 samples
> >
> > | Method\NFE  | 5          | 6          | 8          | 10         | 12         | 15         | 20         |
> > | ----------- | ---------- | ---------- | ---------- | ---------- | ---------- | ---------- | ---------- |
> > | DDIM        | 0.0675     | 0.0530     | 0.0370     | 0.0276     | 0.0216     | 0.0158     | 0.0108     |
> > | +PFDiff-1   | 0.5397     | 0.3172     | 0.0334     | 0.0167     | 0.0111     | 0.0066     | 0.0035     |
> > | +PFDiff-2_1 | **0.0187** | **0.0131** | *0.0090*   | *0.0065*   | *0.0054*   | *0.0040*   | *0.0028*   |
> > | +PFDiff-2_2 | 0.8693     | 0.7190     | 0.2373     | 0.0380     | 0.0232     | 0.0140     | 0.0072     |
> > | +PFDiff-3_1 | *0.0237*   | *0.0197*   | 0.0137     | 0.0110     | 0.0085     | 0.0062     | 0.0041     |
> > | +PFDiff-3_2 | 0.2081     | 0.0399     | **0.0085** | **0.0052** | **0.0037** | **0.0024** | **0.0015** |
> > | +PFDiff-3_3 | 0.9905     | 0.9136     | 0.4091     | 0.0718     | 0.0347     | 0.0208     | 0.0110     |
> >
> > Table A6: LSUN Bedroom 256$\times$256, FID$\downarrow$, 50k samples
> >
> > | Method\NFE  | 5         | 6         | 8         | 10       | 12       | 15       | 20       |
> > | ----------- | --------- | --------- | --------- | -------- | -------- | -------- | -------- |
> > | DDIM        | 71.02     | 47.40     | 26.73     | 19.26    | 15.23    | 11.68    | 9.26     |
> > | +PFDiff-1   | 242.74    | 172.33    | 94.23     | 67.27    | 52.72    | 38.62    | 26.25    |
> > | +PFDiff-2_1 | 78.57     | **18.72** | **11.50** | **9.28** | **8.36** | **7.76** | **7.14** |
> > | +PFDiff-2_2 | 287.01    | 244.58    | 131.94    | 74.24    | 57.70    | 43.93    | 31.28    |
> > | +PFDiff-3_1 | **25.32** | 19.07     | 13.59     | 10.67    | 9.18     | 8.03     | **7.14** |
> > | +PFDiff-3_2 | 221.82    | 114.91    | 12.35     | 9.55     | 8.82     | 8.34     | 7.70     |
> > | +PFDiff-3_3 | 317.00    | 279.26    | 142.31    | 82.88    | 60.68    | 46.47    | 33.17    |

---

> > > ### Comment · Reviewer_7KLd · 2024-11-25
> > >
> > > Although the measured truncation error does not perfectly align with the optimal FID, the automatic search strategy still yields sub-optimal FID, so I increased my rating.

---

> ### Author Response · Authors · 2024-11-25
>
> Thank you for recognizing the automatic search strategy and for increasing the rating!
>
> We also greatly appreciate the reviewer’s valuable and insightful feedback during the review process, which has greatly contributed to the improvement of our manuscript!

---

### Official Review · Reviewer_BQ25 · 2024-11-03

**Soundness:** 4
**Presentation:** 3
**Contribution:** 3
**Rating:** 6
**Confidence:** 4

**Summary:**

The PFDiff paper introduces a novel, training-free, and orthogonal timestep-skipping mechanism to improve existing ODE solvers used in Diffusion Probabilistic Models. The proposed approach helps to reach solutions with fewer NFE, with the aid of springboard along with foresight updates. This addresses a significant challenge in reducing computational cost while keeping high sample quality. Furthermore,  PFDiff improves the efficiency and quality of diffusion model samplig.

**Strengths:**

1). The paper is well written and easy to understand

2). The given illustrations, and provided Algorithms further helps understanding the paper.

3). The paper identifies a limitation of DPMs, which is their sampling efficiency is low as they often require multiple number of denoising steps. Existing methods tend to amplify discretization errors when NFE is below 10, often leading to convergence issues. The proposed approach, named PFDiff, is a training-free and orthogonal timestep-skipping algorithm that helps mitigating these errors while operating with fewer NFEs

4). PFDiff employed the potential for improvements in existing training free accelerated methods, and the sequence of observations that led to the development of PFDiff is remarkable.

5). The proposed sampler can be integrated to any order of ODE solvers regardless of the type.

**Weaknesses:**

I believe this is a good paper as it provide valuable insights while providing solid reasoning, but I have some questions regarding the scalability, as well as about k and h values of the proposed method.

1). I would like to know how will PFDiff maintain quality across different types of diffusion models other than those mentioned in paper?

2). As the algorithm's construction is based on gradients, I would like to know what happens if gradients show a dispersion. How this kind of a scenario is handled? Also is there a possibility of accumulating errors in the proposed approach?

3). A more ablation on the parameters k and h will further enhance the paper. For instance, is it possible to further increase the value of k? At that kind of instance, how would PFDiff work?

**Questions:**

Please see the weaknesses section.

---

> ### Author Response · Authors · 2024-11-17
>
> We sincerely appreciate the reviewer's valuable review of the manuscript and the recognition of our work of the work presented in the paper. Below are our responses to all questions. We kindly hope you could consider increasing the score if you are satisfied.
>
> ***W1: How will PFDiff maintain quality across different types of diffusion models?***
>
> **A:** All diffusion models can be unified under the perspective of SDE/ODE modeling [R1], and the sampling process of PFDiff is built upon the theoretical foundation of ODE. Therefore, theoretically, PFDiff can be applied to all types of diffusion models. Additionally, beyond the DDPM and ScoreSDE [R1] framework models mentioned in the paper, we have also included experimental results from a more advanced diffusion model within the EDM [R2] framework, as shown in the table below:
>
> ImageNet 64$\times$64, FID$\downarrow$
>
> | Method\NFE  | 4         | 6         | 8        | 10       | 12       | 15       | 20       |
> | ----------- | --------- | --------- | -------- | -------- | -------- | -------- | -------- |
> | DDIM        | 58.43     | 34.03     | 22.59    | 16.72    | 13.14    | 10.04    | 7.42     |
> | DDIM+PFDiff | **55.63** | **17.82** | **8.00** | **5.38** | **4.36** | **3.72** | **3.35** |
>
> As can be seen from the table, PFDiff also effectively improves the sampling quality of DDIM within the EDM framework, further validating the broad adaptability of PFDiff across different types of diffusion models.
>
> ***W2.1: How does the algorithm handle scenarios where gradients exhibit dispersion?***
>
> **A:** In fact, the primary design goal of PFDiff is to address the gradient dispersion issue observed in baseline fast solvers (e.g., DDIM). Gradients represent the direction of sampling updates, and when the NFE is small, **dispersion arises from misalignment between the intermediate states that need to be updated and the current direction of the sampling update** (i.e., neural network output). PFDiff mitigates this issue by introducing the foresight update mechanism of Nesterov momentum, which predicts the future sampling update direction. This effectively alleviates the misalignment (i.e., gradient dispersion) between the gradients and the intermediate states that need to be updated in baseline fast solvers, as theoretically guaranteed in Proposition 3.1.
>
> ***W2.2: Is there a possibility of accumulating errors in the proposed approach?***
>
> **A:** The error accumulation in PFDiff arises from the baseline fast solvers (e.g., DDIM). Notably, PFDiff accelerates sampling by reducing the discretization error of the existing fast solvers. Therefore, the error accumulation in PFDiff will be smaller than that in the baseline fast solvers.
>
> ***W3: Can the value of $k$ be further increased, and how would PFDiff perform in that case?***
>
> **A:** Thank you for the reviewer’s suggestion. We have increased the value of the parameter $k$ to 4 on the CIFAR10 dataset and conducted further performance analysis, as shown in the following table:
>
> CIFAR10, FID$\downarrow$
>
> | Method\NFE         | 4         | 6        | 8        | 10       | 12       | 15       | 20       |
> | ------------------ | --------- | -------- | -------- | -------- | -------- | -------- | -------- |
> | DDIM               | 65.70     | 29.68    | 18.45    | 13.66    | 11.01    | 8.80     | 7.04     |
> | +PFDiff($k \le 3$) | **22.38** | 9.84     | **5.64** | 4.57     | 4.39     | 4.10     | **3.68** |
> | +PFDiff($k=4$)     | 27.20     | **8.65** | 6.53     | **4.56** | **4.17** | **3.98** | 3.72     |
>
> From the table, it can be seen that when the value of $k$ is increased to 4, PFDiff improves the sampling quality of DDIM at certain NFE. However, increasing $k$ leads to the problem of over-searching. Therefore, $k \le 3$ might be a more suitable choice, as it provides a better balance between sampling quality and the cost of parameter search.
>
> [R1] Yang Song et al., "Score-based generative modeling through stochastic differential equations." ICLR 2021.
>
> [R2] Tero Karras et al., "Elucidating the design space of diffusion-based generative models." NeurIPS 2022.

---

> ### Author Response · Authors · 2024-11-24
>
> Dear Reviewer BQ25,
>
> Thank you once again for your valuable review of the manuscript and for your recognition of our work! As the discussion period draws to a close, we would appreciate it if you could let us know whether your concerns have been addressed. We would be very pleased to continue the discussion if you have any further questions!
>
> Best, Authors

---

> > ### Comment · Reviewer_BQ25 · 2024-11-25
> >
> > Thank you very much for your detailed response, and my concerns have been addressed. I keep my score unchanged.

---

> ### Author Response · Authors · 2024-11-26
>
> We are so glad to hear that your concerns have been addressed! Thank you once again for recognizing our work!

---

### Official Review · Reviewer_sz2Y · 2024-11-04

**Soundness:** 2
**Presentation:** 3
**Contribution:** 3
**Rating:** 6
**Confidence:** 3

**Summary:**

This paper introduces a novel methodology to accelerate diffusion model sampling. The core concept involves reusing past score predictions to generate a preliminary estimate (springboard) for the next step. Then, future score prediction is obtained from this springboard. By leveraging this future score prediction, the method enables step skipping, directly calculating the point two steps ahead from the current position. This approach offers practical advantages as it is orthogonal to existing advanced samplers and does not require additional training. Extensive experiments demonstrate its effectiveness in significantly accelerating diffusion model sampling when integrated with various state-of-the-art samplers.

**Strengths:**

1. Extensive experiments conducted with diverse models and baselines underscore both the superiority and generality of the proposed methodology. Comprehensive results reveal a substantial improvement in the efficiency of diffusion sampling.
2. By approaching diffusion model acceleration through time-skipping, the authors introduce a technique that is orthogonal to existing advanced samplers. This characteristic, coupled with its training-free nature, enhances its practical applicability.
3. Despite its simplicity and ease of implementation, the methodology presented in the paper yields significant benefits.

**Weaknesses:**

1. The multi-step solver's exclusion of future gradients, a core component of the proposed methodology, undermines the claimed orthogonality. Additionally, the absence of experimental results (Stable diffusion) integrating the method with the DPM-Solver series raises doubts about its performance enhancement potential and the extent of its orthogonality when applied to multi-step solvers.
2.  While the methodology is presented as an orthogonal wrapper for arbitrary ODE solvers, its classification as a standalone ODE solver is also plausible, depending on the perspective.
3. The use of "gradient guidance" in the title and text is potentially misleading. In the context of diffusion models, this term is typically associated with guiding the sampling process using external model gradients (e.g. classifier guidance). For better clarity, using terms like "score" or "predicted noise" would be more appropriate.
4. The direct comparison between the future gradient and the springboard in Figure 2(b) is questionable. Given their different scales, a direct MSE comparison might not be the most accurate approach to assess their relative reliability.

**Questions:**

1. In the Stable diffusion experiment, why was the proposed methodology not applied to DPM-Solver? If the results were presented in the paper, please provide a reference.
2. In Equation 14, is it correct to plug the $n$ points obtained from the $\Delta t$ interval ODE solver into the $2\Delta t$ interval ODE solver? Do I understand it correctly?
3. Are the MSE scales of the future gradient and the springboard directly comparable? Would the author(s) think that using the MSE of the image updated with the future gradient instead of the future gradient in Figure 2(b) provides a more meaningful comparison?
4. Is the mention of Nesterov momentum solely due to the similarity in form between the proposed springboard prediction method and Nesterov momentum? Have any properties of Nesterov momentum, such as improved convergence speed, been leveraged in the theoretical analysis or practical implementation of the proposed method?

---

> ### Author Response · Authors · 2024-11-17
>
> We sincerely appreciate the reviewer's meticulous review and insightful comments, which have helped improve our paper. Below are our responses to all the questions. Please consider increasing the score if you are satisfied.
>
> ***W1.1: The multi-step solver undermines the orthogonality.***
>
> **A:** It is important to emphasize that, in theory, higher-order (not multi-step) solvers and future gradients are orthogonal, as shown in Equation (14). However, in the practical experiments, we found that introducing future gradients into higher-order solvers is highly sensitive to the choices of hyperparameters $k$ and $h$, leading to unstable results. Removing the future gradients and only adding past gradients (which is also the core component of our method) to the higher-order solver yields more stable performance, which significantly mitigates the convergence issues, as shown in Figure 3(c) (originally Figure 4(c)).
>
> ***W1.2 and Q1: Why are there no experimental results applying PFDiff to DPM-Solver in Stable Diffusion?***
>
> **A:** We did not include the experimental results for the following reasons: First, DDIM and DPM-Solver-1 are equivalent [R1]. Second, with fewer NFE, higher-order DPM-Solvers perform worse than DDIM on Stable-Diffusion [R2]. Finally, DDIM+PFDiff has outperformed other previously best-performing solvers. Additionally, we also provide experimental results applying PFDiff to DPM-Solver in Stable Diffusion with a guidance scale of 7.5, as shown in the table below:
>
> Stable Diffusion, FID$\downarrow$
>
> | Method\NFE          | 4         | 6         | 8         | 10        | 15        | 20        |
> | ------------------- | --------- | --------- | --------- | --------- | --------- | --------- |
> | DPM-Solver-1 [R1]   | 35.48     | 20.33     | 17.46     | 16.78     | 16.08     | 15.95     |
> | DPM-Solver-1+PFDiff | **29.02** | **15.47** | **13.26** | **13.06** | **13.57** | **13.97** |
> | DPM-Solver-2 [R1]   | 184.21    | 157.95    | 148.67    | 135.81    | 92.62     | 40.47     |
> | DPM-Solver-2+PFDiff | 147.20    | 106.24    | 57.07     | 31.66     | 17.87     | 14.13     |
>
> As shown in the table above, PFDiff further improves the sampling quality of DPM-Solver-1 and DPM-Solver-2. Notably, the results in the table come from the original DPM-Solver implementation, while the DPM-Solver results in the paper utilized additional tricks from [R2] to achieve better performance.
>
> ***W2: The methodology can be classified as a standalone ODE solver.***
>
> **A:** We strongly agree with the insightful classification: the methodology can be classified as a standalone ODE solver, depending on the perspective. We have incorporated this classification in Section 3.4 (line 337 to line 339).
>
> ***W3: The use of "gradient guidance" in the title and text is potentially misleading.***
>
> **A:** We appreciate the reviewer for pointing out the potential misunderstanding in the title. As revised, the title now reads: "PFDiff: Training-Free Acceleration of Diffusion Models Combining Past and Future Scores." Additionally, we have made corresponding changes in the paper, replacing "gradient" with "score" and removing the expression of "gradient guidance".
>
> ***W4 and Q3: Are the MSE scales of the future gradient and the springboard directly comparable?***
>
> **A:** Thank you for raising this very valuable question. Based on the suggestion in Q3, we have added a fairer comparison in the paper, showing the MSE of the status separately updated using the "springboard" and future gradient. The corresponding results are presented in Figure 6(b) in Appendix D.8. Notably, the trends in Figure 6(b) and Figure 1(b) (original Figure 2(b)) are consistent, which validates the effectiveness of using the future gradient.
>
> ***Q2: In Equation 14, is it correct to plug the $n$ points obtained from the $\Delta$ interval ODE solver into the $2\Delta$ interval ODE solver?***
>
> **A:** Yes, you are nearly correct. Equation 14 represents plugging the $p$ (not $n$) points obtained from the $\Delta$ interval ODE solver into the $2\Delta$ interval ODE solver. However, for first-order ODE solvers (e.g., DDIM), $p=1$. This means that we use the gradient corresponding to the endpoint $t_{i+1}$ of the interval $\Delta$ as a replacement for the gradient corresponding to the endpoint $t_{i}$ of the interval $2\Delta$. This replacement effectively reduces the discretization error of the first-order ODE solver, and a detailed theoretical analysis can be found in Proposition 3.1.
>
> [R1] Cheng Lu et al., "DPM-solver: a fast ODE solver for diffusion probabilistic model sampling in around 10 steps." NeurIPS 2022.
>
> [R2] Cheng Lu et al., "Dpm-solver++: Fast solver for guided sampling of diffusion probabilistic models." arXiv:2211.01095.

---

> ### Author Response · Authors · 2024-11-17
>
> ***Q4: Is there a theoretical analysis of Nesterov momentum on convergence speed?***
>
> **A:** In this work, we only discuss Nesterov momentum as the motivation for using future gradients. Nevertheless, in Equation (15) and Proposition 3.1, we employ a Taylor expansion and further derivations to theoretically demonstrate that the use of future gradients can reduce discretization errors, thereby accelerating the convergence of the sampling process. This theoretical result indirectly suggests that the foresight update mechanism of Nesterov momentum can effectively accelerate convergence.

---

> ### Author Response · Authors · 2024-11-24
>
> Dear Reviewer sz2Y,
>
> Thank you once again for your insightful feedback! These suggestions have greatly enhanced the quality of our manuscript. As the discussion period draws to a close, we would appreciate it if you could let us know whether your concerns have been addressed. We would be very pleased to continue the discussion if you have any further questions!
>
> Best, Authors

---

> > ### Comment · Reviewer_sz2Y · 2024-11-25
> >
> > Thanks to the authors for addressing most of my concerns. I've increased the final score accordingly.
> >
> > To further assert orthogonality, I suggest adding a new table for W1.2 to the main text. Demonstrating PFDiff's improvement across multiple solvers would be ideal.
> >
> > For the final version, I'd prefer Figure 6(b) over Figure 1(b) as the comparison within the same image domain is more intuitive.
> >
> > Thanks again for the authors' efforts.

---

> ### Author Response · Authors · 2024-11-25
>
> Thank you very much for your response and for increasing the rating! We are so glad to hear that most of your concerns have been addressed!
>
> In response to the reviewer’s suggestions, we have revised the relevant content and uploaded the revised paper. Specifically, we added Table 1 (i.e., the W1.2 table) to Section 4.2 to demonstrate the orthogonality of PFDiff. Meanwhile, we replaced Figure 1(b) with Figure 6(b) to provide a more intuitive comparison within the same image domain. Furthermore, we also added and modified the corresponding descriptions in Section 4.2 and Appendix D.8.
>
> Thank you once again for your valuable and insightful feedback, which has greatly improved the quality of our manuscript.

---

### Author Response · Authors · 2024-11-17

We sincerely appreciate the effort of all the reviewers for their detailed review and insightful suggestions. We would like to present the following modifications to the paper (the revised version has been uploaded, and the changes are highlighted in blue within the document):

- We have revised the title and aligned the relevant expressions in the paper, including replacing "gradient" with "score" and removing the expression of "gradient guidance", to eliminate potential misinterpretations (Reviewer sz2Y, W3).
- We have added Appendix D.8, which provides additional experimental details and results for Figures 1(a) and (b) (original Figures 2(a) and (b)). (Reviewer sz2Y, W4/Q3; Reviewer 7KLd, W1)
- Section 3.5 has been added to explain the effectiveness of PFDiff through the geometric shape of the trajectories. (Reviewer 7KLd, W1)
- An insightful comment regarding PFDiff has been added to lines 337–339 of Section 3.4. (Reviewer sz2Y, W2)
- Several writing expressions have been revised, specifically in lines 68-69, 183–184, and 327–328. (Reviewer 7KLd, W2)
- We have moved Figure 1 to Appendix D.9 (original Appendix D.8) and corrected the citation years for some references. (Reviewer 7KLd, Others)
- We added Table 1 in Section 4.2 to validate the orthogonality of PFDiff in Stable Diffusion and also revised the relevant descriptions in Section 4.2 and Appendix D.5. (Reviewer sz2Y, W1.2/Q1; Reviewer 7KLd, W6)
- We added a description and experimental results regarding the automatic search strategy for selecting $k$ and $h$ in lines 493-495 of Section 4.3 and in Appendix D.6.1. (Reviewer 7KLd, New question)

---

### Meta-Review · Area_Chair_Ua4h · 2024-12-17

**Metareview:**

The work proposed PFDiff, a training-free, and orthogonal timestep-skipping mechanism that improves existing ODE solvers in Diffusion Probabilistic Models. By leveraging springboard and foresight updates, the method achieves solutions with fewer function evaluations (NFE), significantly reducing computational costs while preserving high sample quality. This approach effectively improves both the efficiency and the quality of diffusion model sampling, addressing a key challenge in the field.

All reviewers find the idea interesting, with strong empirical results.

**Additional Comments On Reviewer Discussion:**

During the rebuttal:
* the authors have revised the title and aligned the relevant expressions in the paper to to eliminate potential misinterpretations of the results.

* The authors provided additional experimental details and results, comparing using different solvers (i.e., DPM) and stable diffusion for more comprehensive evaluations.

---

### Decision · Program_Chairs · 2025-01-22

Accept (Poster)